# PT²-LLM: Post-Training Ternarization for Large Language Models

**Xianglong Yan**[1][*], **Chengzhu Bao**[1][*], **Zhiteng Li**[1], **Tianao Zhang**[1], **Kaicheng Yang**[1],
**Haotong Qin**[2], **Ruobing Xie**[3], **Xingwu Sun**[3], **Yulun Zhang**[1][†]
[1]Shanghai Jiao Tong University, [2]ETH Zürich, [3]Tencent Hunyuan

## Abstract

Large Language Models (LLMs) have shown impressive capabilities across diverse tasks, but their large memory and compute demands hinder deployment. Ternarization has gained attention as a promising compression technique, delivering substantial size reduction and high computational efficiency. However, its potential in the post-training quantization (PTQ) setting remains underexplored, due to the challenge of training-free parameter optimization and the quantization difficulty posed by outliers and dispersed weights. To address these issues, we propose PT²-LLM, a post-training ternarization framework tailored for LLMs. At its core is an Asymmetric Ternary Quantizer equipped with a two-stage refinement pipeline: (1) Iterative Ternary Fitting (ITF), which alternates between optimal ternary grid construction and flexible rounding to minimize quantization error, and (2) Activation-aware Grid Alignment (AGA), which further refines the ternary grid to better match full-precision outputs. In addition, we propose a plug-and-play Structural Similarity-based Reordering (SSR) strategy that leverages inter-column structural similarity to ease quantization and mitigate outlier effects, further enhancing overall performance. Extensive experiments demonstrate that PT²-LLM delivers competitive performance against state-of-the-art (SOTA) 2-bit PTQ methods with lower memory cost, while also accelerating both prefill and decoding to achieve end-to-end speedup. The code and models will be available at https://github.com/XIANGLONGYAN/PT2-LLM.

## 1 Introduction

Large Language Models (LLMs) (Touvron et al., 2023a;b; Dubey et al., 2024; Yang et al., 2025; Zhang et al., 2022) have achieved remarkable progress in language understanding, reasoning, and generation. They serve as the foundation for many real-world applications and remain at the forefront of AI research. However, these achievements are largely enabled by the massive scale of model parameters. Modern LLMs often contain tens or even hundreds of billions of parameters (*e.g.,* DeepSeek-R1 (Guo et al., 2025) has 671 billion), leading to substantial memory consumption and intensive computational demands. Running such models demands powerful GPUs, large memory, and high energy, which hinders deployment on resource-limited or latency-sensitive platforms.

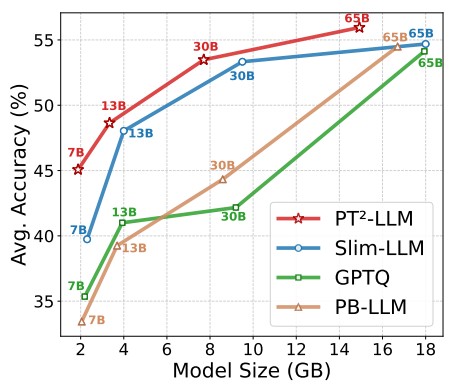

Figure 1: LLaMA performance on 7 zero-shot Question Answering (QA) datasets. PT²-LLM yields the best accuracy at equal memory cost.

Weight-only quantization (Frantar et al., 2023) reduces weight precision to save memory and accelerate inference. Among various schemes, ternarization (Li et al., 2016) constrains weights to $\{-1, 0, +1\}$, enabling high compression ratios and efficient computation. Compared to low-bit quantization (*e.g.,* 2–4 bit) (Lin et al., 2024b), it eliminates most floating-point multiplications by using simple additions, reducing both computational and energy costs. Compared to binarization (Rastegari et al., 2016), ternarization better fits the unimodal distribution of LLM weights and offers stronger representational capacity, yielding higher accuracy. Balancing efficiency and expressiveness, ternarization suits resource-limited LLM deployment (Wang et al., 2025; Yin et al., 2025).

[*]Equal contribution
[†]Corresponding author: Yulun Zhang, yulun100@gmail.com

Recent studies (Lu et al., 2024; Zhang et al., 2020) on ternarization primarily focus on quantization-aware training (QAT), where models are trained under ternary constraints. Such methods are mainly explored on moderate-sized architectures like BERT or DiT, where training remains affordable. While attempts have been made to extend QAT-based ternarization to LLMs (*e.g.,* BitNet b1.58 (Ma et al., 2024)), such approaches are highly impractical due to the immense parameter scale and the prohibitive demands on training resources, computational budget, and full access to training data. In contrast, post-training quantization (PTQ) offers a far more practical and efficient alternative: it enables rapid conversion from full-precision models to compact ternary versions without retraining or access to full training data, making it more suitable for real-world LLM deployment scenarios.

However, PTQ-based ternarization remains underexplored, as its direct application often causes severe performance degradation, making models unusable. Through analysis, we identify two main challenges: **(i)** Unlike QAT, which optimizes ternary parameters through gradient-based updates on large-scale training data, PTQ must efficiently refine them without any training, which poses a core challenge. **(ii)** As an extreme low-bit quantization scheme, ternarization struggles to represent weights with dispersed or outlier-heavy distributions, making it particularly prone to large quantization error.

In this paper, we propose **PT$^2$-LLM**, a post-training ternarization framework tailored for LLMs. To tackle the challenge of training-free ternary parameter optimization, we propose an Asymmetric Ternary Quantizer (ATQ), refined through two stages: Iterative Ternary Fitting (ITF) and Activation-aware Grid Alignment (AGA). ITF alternates between optimal ternary grid construction and flexible rounding to minimize quantization error, while AGA leverages calibration data to further align ternary outputs with full-precision ones. To handle dispersed weights and outliers, we propose a plug-and-play Structural Similarity-based Reordering (SSR) strategy, which reorganizes columns based on inter-column structural correlation to ease quantization. Equipped with ATQ and SSR, **PT$^2$-LLM** enables accurate and robust post-training ternarization. As shown in Fig. 1, it outperforms state-of-the-art (SOTA) 2-bit PTQ methods in zero-shot QA accuracy under the same memory budget.

Our key contributions can be summarized as follows:

- We propose **PT$^2$-LLM**, a novel ternarization framework that efficiently compresses pre-trained LLMs into a ternary grid without any retraining, addressing the unexplored challenges of post-training ternarization in LLMs.

- We design an Asymmetric Ternary Quantizer for post-training ternarization. It is optimized through two training-free stages: Iterative Ternary Fitting (ITF) and Activation-aware Grid Alignment (AGA). These components enable effective refinement of ternary parameters, reducing quantization error and improving alignment with full-precision outputs.

- We develop a plug-and-play Structural Similarity-based Reordering (SSR) strategy. It reorders weight columns based on structural similarity, which helps reduce quantization difficulty and suppress the influence of outliers.

- Extensive experiments demonstrate the competitive performance of **PT$^2$-LLM** compared to SOTA 2-bit PTQ methods, with reduced memory consumption and faster inference.

## 2 RELATED WORKS

### 2.1 NETWORK TERNARIZATION

Ternarization compresses neural networks by constraining parameters to $\{-1, 0, +1\}$, making them memory and computation efficient while preserving strong representational capacity. Ternary weight networks (TWN) (Li et al., 2016) introduced scale-aware ternary quantization to minimize the Euclidean distance from full precision, achieving up to $16\times$ compression. Trained ternary quantization (TTQ) (Zhu et al., 2016) improves accuracy by jointly learning both ternary weight values and their assignments during training. Further works (Wang et al., 2018; Alemdar et al., 2017) extended ternarization to activations, enabling fully ternary networks with greater efficiency. More recently, efforts have aimed to apply ternarization to larger and more complex models. TernaryBERT (Zhang et al., 2020) achieved $14.9\times$ BERT compression via loss-aware ternarization and distillation. TerDiT (Lu et al., 2024) scaled ternarization to 4.2B diffusion transformers. TernaryLLM (Chen et al., 2024) introduced learnable scaling and feature distillation, outperforming prior low-bit LLMs. BitNet b1.58 (Ma et al., 2024) proposed a ternary-weight training framework for LLMs, delivering near full-precision accuracy with reduced latency and energy consumption. However, most existing ternarization methods rely on training, limiting their practicality in real-world deployment.

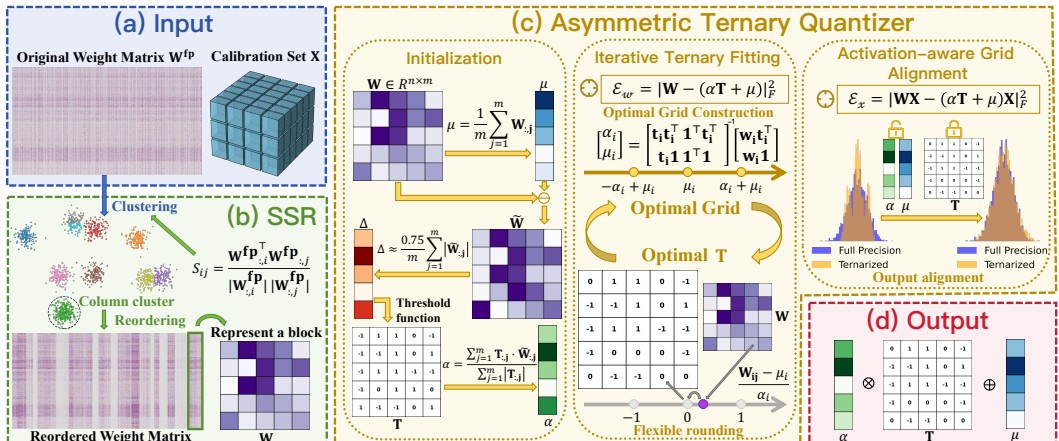

Figure 2: Overview of PT²-LLM. **Structural Similarity-based Reordering (SSR)**: reorders columns based on structural similarity. **Asymmetric Ternary Quantizer**: enhanced by Iterative Ternary Fitting (ITF) and Activation-aware Grid Alignment (AGA) for refined ternary parameter optimization.

## 2.2 QUANTIZATION FOR LARGE LANGUAGE MODELS

Quantization reduces the memory footprint and inference cost of large language models, and is typically categorized into quantization-aware training (QAT) and post-training quantization (PTQ).

**Quantization-Aware Training (QAT).** QAT incorporates quantization during training, enabling LLMs to learn robust low-bit representations through backpropagation. Works such as LLM-QAT (Liu et al., 2024a) and BitDistiller (Du et al., 2024) leverage knowledge distillation to preserve accuracy under low-bit quantization. EfficientQAT (Chen et al., 2025) reduces QAT's overhead via a two-stage training scheme. Recent efforts like Onebit (Xu et al., 2024) and BinaryMoS (Jo et al., 2024) further extend QAT to the 1-bit regime. While QAT can effectively preserve performance under low-bit quantization, its high computational and memory cost remains a major limitation.

**Post-Training Quantization (PTQ).** Unlike QAT, PTQ directly quantizes pretrained models without retraining, making it more efficient and deployment-friendly for LLMs. Early methods (Dettmers et al., 2022; Yao et al., 2022; Li et al., 2021) improve quantization by introducing grouping labels. Techniques like AWQ (Lin et al., 2024b) and OWQ (Lee et al., 2024) introduced transformations that scale salient weights, aiming to preserve activation expressiveness and overall model capacity. GPTQ (Frantar et al., 2023) leverages Hessian-guided error compensation, with GP-TAQ (Li et al., 2025a) extending it via asymmetric calibration. OmniQuant (Shao et al., 2023) and SmoothQuant (Xiao et al., 2023) address activation outliers through scale redistribution. More recent work (Lin et al., 2024a; Ashkboos et al., 2024; Liu et al., 2024b) adopts rotation-based transformations for improved low-bit quantization. For ultra-low-bit settings, QuIP (Chee et al., 2025) and QuIP# (Tseng et al., 2025) use incoherence processing to boost performance, while Slim-LLM (Huang et al., 2025) employs salience-aware mixed-precision schemes. In the binarization domain, 1-bit methods such as PB-LLM (Shang et al., 2024), BiLLM (Huang et al., 2024), and ARB-LLM (Li et al., 2025b) show competitive results. Sub-1-bit approaches (Dong et al., 2025; Yan et al., 2025; Gu et al., 2025) further advance compression by reducing average bitwidths while retaining strong accuracy. Our method falls under the category of post-training ternary quantization.

## 3 METHOD

**Overview.** Fig. 2 illustrates the overall workflow of PT²-LLM. We first review the standard symmetric ternarization formulation and basic notations in Section 3.1. Building on this foundation, Section 3.2 introduces the Asymmetric Ternary Quantizer, which features two training-free stages: Iterative Ternary Fitting (ITF) and Activation-aware Grid Alignment (AGA). Section 3.3 then presents the Structural Similarity-based Reordering (SSR), demonstrating how column clustering by structural similarity can be effectively combined within the GPTQ framework.

## 3.1 PRELIMINARY

**Symmetric Ternarization.** Symmetric ternarization compresses full-precision weights into the ternary set $\{-1, 0, +1\}$ by minimizing the discrepancy between the original weight matrix and its

ternary approximation under an appropriate scaling:

$$\alpha^*, \ \mathbf{T}^* = \arg\min_{\alpha, \ \mathbf{T}} \|\mathbf{W} - \alpha\mathbf{T}\|_F^2, \tag{1}$$

where $\mathbf{W} \in \mathbb{R}^{n \times m}$ is the full-precision weight matrix, $\alpha \in \mathbb{R}^{n \times 1}$ is a row-wise scaling factor, and $\mathbf{T} \in \{-1, 0, +1\}^{n \times m}$ is the ternary matrix. Since jointly optimizing $\alpha$ and $\mathbf{T}$ causes parameter coupling, TWN (Li et al., 2016) proposes a threshold-based solution. Specifically, for each element $\mathbf{W}_{ij}$, a row-wise threshold $\Delta \in \mathbb{R}^{n \times 1}$ is used to determine the corresponding ternary value $\mathbf{T}_{ij}$ as:

$$\mathbf{T}_{ij} = \begin{cases} 1, & \text{if } \mathbf{W}_{ij} > \Delta_i, \\ 0, & \text{if } |\mathbf{W}_{ij}| \leq \Delta_i, \\ -1, & \text{if } \mathbf{W}_{ij} < -\Delta_i. \end{cases} \tag{2}$$

Given a fixed threshold $\Delta$, the ternary matrix $\mathbf{T}$ is deterministically defined, enabling a closed-form solution for the optimal scaling factor $\alpha$. Since directly optimizing $\Delta$ is difficult in practice, TWN approximates $\Delta$ based on assumed weight distributions. Assuming uniform or normal priors, $\Delta$ is approximated by a scaled mean of absolute weights, and the optimal $\alpha$ follows:

$$\Delta \approx \frac{0.75}{m} \sum_{j=1}^{m} |\mathbf{W}_{:,j}|, \quad \alpha = \frac{\sum_{j=1}^{m} \mathbf{T}_{:,j} \cdot \mathbf{W}_{:,j}}{\sum_{j=1}^{m} |\mathbf{T}_{:,j}|}. \tag{3}$$

This approximation enables fast and training-free ternarization by decoupling $\alpha$ and $\mathbf{T}$, providing a practical solution to ternary parameters initialization in PTQ settings.

## 3.2 Asymmetric Ternary Quantizer

**Asymmetric Ternary Initialization.** Empirical observations reveal that the weight distributions in LLMs are not always symmetric, as many layers exhibit non-zero means. We provide visualizations in the supplementary file to further support this observation. While symmetric ternarization (as discussed in Section 3.1) performs well under QAT due to its ability to reshape the weight distribution through backpropagation, this assumption no longer holds in PTQ, where pre-trained weights remain fixed. To better capture the bias in pre-trained weights, we follow prior work (Chen et al., 2024) and adopt an asymmetric ternarization scheme by introducing a row-wise offset $\mu \in \mathbb{R}^{n \times 1}$, initialized as the mean of each row. The dequantized weight $\widehat{\mathbf{W}}$ is then computed as:

$$\widehat{\mathbf{W}} = \alpha\mathbf{T} + \mu, \quad \mu = \frac{1}{m} \sum_{j=1}^{m} \mathbf{W}_{:,j}. \tag{4}$$

For the initialization of $\alpha$ and the ternary matrix $\mathbf{T}$, we follow the same strategy described in Section 3.1, applying it to the centered weight matrix $\widetilde{\mathbf{W}} = \mathbf{W} - \mu$ to remove bias:

$$\Delta \approx \frac{0.75}{m} \sum_{j=1}^{m} |\widetilde{\mathbf{W}}_{:,j}|, \quad \alpha = \frac{\sum_{j=1}^{m} \mathbf{T}_{:,j} \cdot \widetilde{\mathbf{W}}_{:,j}}{\sum_{j=1}^{m} |\mathbf{T}_{:,j}|}. \tag{5}$$

$\mathbf{T}$ is still initialized using Eq. 2, with $\Delta$ applied to $\widetilde{\mathbf{W}}$. This asymmetric initialization offers a stable and expressive foundation for post-training ternarization under non-zero-mean weight distributions.

**Iterative Ternary Fitting.** After initialization, we obtain the three key components of ternarization: the scaling factor $\alpha$, the shift parameter $\mu$, and the ternary matrix $\mathbf{T}$. $\alpha$ and $\mu$ together define a ternary grid with only three possible quantized values for each row $i$, namely $\{-\alpha_i + \mu_i, \ \mu_i, \ \alpha_i + \mu_i\}$. How to refine this ternary grid so that it better fits the underlying weight distribution is crucial for improving quantization quality. We first define the quantization error of weights $\mathcal{E}_w$ as:

$$\mathcal{E}_w = \|\mathbf{W} - \widehat{\mathbf{W}}\|_F^2, \quad \text{where } \widehat{\mathbf{W}} = \alpha\mathbf{T} + \mu. \tag{6}$$

Our current optimization objective is to minimize the quantization error $\mathcal{E}_w$, which is achieved by optimizing the ternarization parameters $\alpha$, $\mu$, and $\mathbf{T}$. Since $\alpha$ and $\mu$ together determine the discrete grid values of ternarization, we refer to them as the ternary grid parameters. A well-constructed grid is essential to provide a reliable basis for subsequent optimization of the ternary matrix $\mathbf{T}$. Therefore, we first focus on establishing a high-quality ternary grid. By differentiating the quantization error $\mathcal{E}_w$ with respect to $\alpha_i$ and $\mu_i$, we obtain the following gradients:

$$\frac{\partial \mathcal{E}_w}{\partial \alpha_i} = 2\big(\alpha_i \mathbf{t}_i + \mu_i \mathbf{1}^\top - \mathbf{w}_i\big)\mathbf{t}_i^\top, \quad \frac{\partial \mathcal{E}_w}{\partial \mu_i} = 2\big(\alpha_i \mathbf{t}_i + \mu_i \mathbf{1}^\top - \mathbf{w}_i\big)\mathbf{1}, \tag{7}$$

where $\mathbf{t}_i \in \mathbb{R}^{1 \times m}$ denotes the $i$-th row of $\mathbf{T}$, and $\mathbf{w}_i \in \mathbb{R}^{1 \times m}$ is the corresponding row of $\mathbf{W}$. $\mathbf{1} \in \mathbb{R}^{m \times 1}$ is a column vector of all ones. The parameters $\alpha_i$ and $\mu_i$ serve as the scaling factor and

shift associated with the $i$-th row, respectively. To obtain the optimal ternary grid parameters, we set the partial derivatives to zero and solve for $\alpha_i$ and $\mu_i$. This leads to a system of linear equations for the optimal grid parameters $\alpha_i$ and $\mu_i$ (see supplementary file for detailed derivation):

$$\frac{\partial \mathcal{E}_w}{\partial \alpha_i} = 0, \quad \frac{\partial \mathcal{E}_w}{\partial \mu_i} = 0 \implies \begin{bmatrix} \mathbf{t}_i \mathbf{t}_i^\top & \mathbf{1}^\top \mathbf{t}_i^\top \\ \mathbf{t}_i \mathbf{1} & \mathbf{1}^\top \mathbf{1} \end{bmatrix} \begin{bmatrix} \alpha_i \\ \mu_i \end{bmatrix} = \begin{bmatrix} \mathbf{w}_i \mathbf{t}_i^\top \\ \mathbf{w}_i \mathbf{1} \end{bmatrix}, \tag{8}$$

which can be efficiently solved to obtain the optimal ternary grid for the $i$-th row. To enable efficient batched computation across rows, we further reformulate the optimal solutions of $\alpha^*$ and $\mu^*$ into a more compact vectorized form (see supplementary file for detailed derivation):

$$\alpha^* = \frac{m \cdot (\mathbf{W} \circ \mathbf{T})\mathbf{1} - (\mathbf{T}\mathbf{1}) \circ (\mathbf{W}\mathbf{1})}{m \cdot (\mathbf{T} \circ \mathbf{T})\mathbf{1} - (\mathbf{T}\mathbf{1})^2}, \quad \mu^* = \frac{(\mathbf{T} \circ \mathbf{T})\mathbf{1} \circ (\mathbf{W}\mathbf{1}) - (\mathbf{T}\mathbf{1}) \circ [(\mathbf{W} \circ \mathbf{T})\mathbf{1}]}{m \cdot (\mathbf{T} \circ \mathbf{T})\mathbf{1} - (\mathbf{T}\mathbf{1})^2}, \tag{9}$$

where $\circ$ denotes element-wise multiplication, with all divisions also element-wise. $m$ is the number of elements per row. This vectorized form enables parallel closed-form solutions across rows, ensuring optimal $\alpha^*$ and $\mu^*$ under fixed $\mathbf{T}$ and thus the best ternary grid at the current stage. After obtaining the current optimal ternary grid, we update $\mathbf{T}$ by mapping the full-precision weights onto it. Instead of a fixed threshold, which is rigid and often suboptimal for diverse weight distributions, we adopt a more flexible element-wise ternary rounding to minimize the quantization error $\mathcal{E}_w$ (Eq. 6). Given $\alpha^*$ and $\mu^*$, the optimal value of each entry $\mathbf{T}_{ij}^*$ is determined by the following rule:

$$\mathbf{T}_{ij}^* = \arg\min_{t \in \{-1,0,1\}} |\mathbf{Z}_{ij} - t|, \quad \text{where } \mathbf{Z}_{ij} = \frac{\mathbf{W}_{ij} - \mu_i^*}{\alpha_i^*}. \tag{10}$$

This guarantees that, under fixed $\alpha^*$ and $\mu^*$, the updated $\mathbf{T}^*$ yields the minimal quantization error $\mathcal{E}_w$, making it the optimal ternary assignment for the current grid. We observe that obtaining the optimal ternary grid and the optimal ternary matrix naturally forms an iterative optimization scheme. By alternating between Eq. 9 and Eq. 10, the algorithm greedily reduces the quantization error $\mathcal{E}_w$ at each step. Convergence is reached when the update in Eq. 10 no longer changes the ternary matrix $\mathbf{T}$, indicating that the ternarized structure has stabilized. In practice, it converges within about 10 iterations.

**Algorithm 1** Pseudocode of the Asymmetric Ternary Quantizer. See supp. file for details.

---
func ATQ($\mathbf{W}, \mathbf{X}$)
**Input:** $\mathbf{W} \in \mathbb{R}^{n \times m}$ - weight matrix
$\quad\quad\quad \mathbf{X} \in \mathbb{R}^{B \times L \times m}$ - calibration data
**Output:** $\widehat{\mathbf{W}} \in \mathbb{R}^{n \times m}$
1: $\alpha, \mu, \mathbf{T} := \text{Ternary\_Init}(\mathbf{W})$
2: $\mathbf{T}_{prev} := \mathbf{0}$
3: **while** $\mathbf{T} \neq \mathbf{T}_{prev}$ **do**
4: $\quad \mathbf{T}_{prev} \leftarrow \mathbf{T}$
5: $\quad \alpha, \mu \leftarrow \text{Biuld\_Optimal\_Grid}(\mathbf{T}, \mathbf{W})$
6: $\quad \mathbf{T} \leftarrow \text{Flexible\_Round}(\mathbf{W}, \alpha, \mu)$
7: **end while**
8: $\alpha, \mu \leftarrow \text{AGA}(\mathbf{W}, \mathbf{T}, \mathbf{X})$
9: $\widehat{\mathbf{W}} \leftarrow \alpha\mathbf{T} + \mu$
10: **return** $\widehat{\mathbf{W}}$

---

**Activation-aware Grid Alignment.** While Iterative Ternary Fitting effectively minimizes the weight quantization error $\mathcal{E}_w$, the actual output of LLMs depends on the interaction between weights and activations. To address this issue, we introduce the activation-aware output error $\mathcal{E}_x$:

$$\mathcal{E}_x = \|\mathbf{W}\mathbf{X} - \widehat{\mathbf{W}}\mathbf{X}\|_F^2, \quad \text{where } \widehat{\mathbf{W}} = \alpha\mathbf{T} + \mu. \tag{11}$$

Here, $\mathbf{X} \in \mathbb{R}^{B \times L \times m}$ denotes the calibration data with batch size $B$, sequence length $L$, and embedding dimension $m$. This formulation directly couples quantization with the model outputs, ensuring that the optimization better reflects the real scenario. In line with the Iterative Ternary Fitting, we again differentiate $\mathcal{E}_x$ with respect to $\alpha_i$ and $\mu_i$, set the derivatives to zero, and obtain a system of equations that gives the optimal ternary grid under the current objective. As before, the solution is expressed row-wise for the $i$-th component:

$$\frac{\partial \mathcal{E}_x}{\partial \alpha_i} = 0, \quad \frac{\partial \mathcal{E}_x}{\partial \mu_i} = 0 \implies \begin{bmatrix} \mathbf{t}_i \mathbf{C} \mathbf{t}_i^\top & \mathbf{1}^\top \mathbf{C} \mathbf{t}_i^\top \\ \mathbf{t}_i \mathbf{C} \mathbf{1} & \mathbf{1}^\top \mathbf{C} \mathbf{1} \end{bmatrix} \begin{bmatrix} \alpha_i \\ \mu_i \end{bmatrix} = \begin{bmatrix} \mathbf{w}_i \mathbf{C} \mathbf{t}_i \\ \mathbf{w}_i \mathbf{C} \mathbf{1} \end{bmatrix}, \tag{12}$$

where $\mathbf{C} = \sum_b \sum_i \mathbf{X}_{bi} \mathbf{X}_{bi}^\top$. By solving this system, we obtain the closed-form solutions:

$$\alpha^* = \frac{d \cdot (\mathbf{W} \circ \mathbf{T})\mathbf{S}\mathbf{1} - \mathbf{v} \circ (\mathbf{W}\mathbf{S}\mathbf{1})}{d \cdot \mathbf{T}^2 \mathbf{S}\mathbf{1} - \mathbf{v}^2}, \quad \mu^* = \frac{\mathbf{T}^2 \mathbf{S}\mathbf{1} \circ (\mathbf{W}\mathbf{S}\mathbf{1}) - \mathbf{v} \circ [(\mathbf{W} \circ \mathbf{T})\mathbf{S}\mathbf{1}]}{d \cdot \mathbf{T}^2 \mathbf{S}\mathbf{1} - \mathbf{v}^2}, \tag{13}$$

where $d = \mathbf{1}^\top \mathbf{S}\mathbf{1}$ is a scalar, and $\mathbf{v} = \mathbf{T}\mathbf{S}\mathbf{1}$; $\mathbf{T}^2$ and $\mathbf{v}^2$ denote element-wise squares. This activation-aware alignment of grid parameters significantly improves the consistency between quantized and full-precision outputs. Ideally, we would update $\mathbf{T}$ to further reduce the output error $\mathcal{E}_x$, but unlike ITF, no optimal solution exists. Greedy search is possible, yet in practice we observe that updating $\mathbf{T}$ leads to severe overfitting on the calibration set. Therefore, we freeze $\mathbf{T}$ and update only $(\alpha, \mu)$ once, which already yields accurate approximations. See supplementary file for details on overfitting.

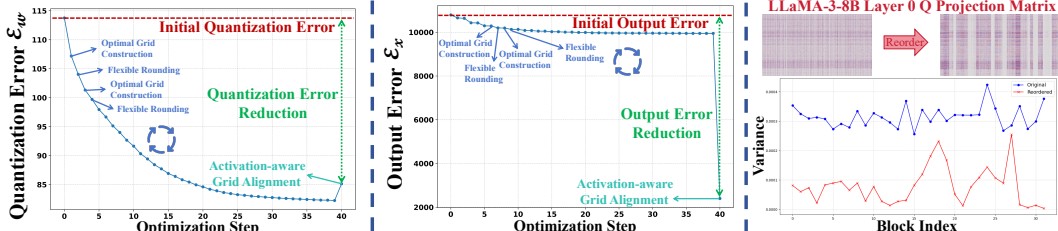

Figure 3: Visualization of the proposed Asymmetric Ternary Quantizer (ATQ) and Structural Similarity-based Reordering (SSR) effects. **Left:** Quantization error $\mathcal{E}_w$ across optimization steps during ATQ. **Middle:** Output error $\mathcal{E}_x$ across optimization steps during ATQ. **Right:** After column reordering, the block-wise variance becomes smaller, showing a more compact weight distribution.

**Overall ATQ Workflow.** As shown in Algorithm 1, ATQ refines the ternary parameters via ITF for a more accurate $\mathbf{T}$, and then applies AGA to further align the ternary grid parameters with the output, yielding the final quantized weights. In Fig. 3 (left and middle), we plot quantization error $\mathcal{E}_w$ and output error $\mathcal{E}_x$ across steps. $\mathcal{E}_w$ steadily decreases during ITF and slightly rises after the AGA update as the optimization objective shifts, while $\mathcal{E}_x$ drops modestly during ITF and sharply after AGA. Overall, the results show ATQ reduces both quantization and output errors without retraining.

### 3.3 STRUCTURAL SIMILARITY-BASED REORDERING

**Motivation.** Following GPTQ (Frantar et al., 2023), our ternarization is blockwise: large weight matrices are split into fixed-size blocks and quantized independently. While this improves accuracy over quantizing the entire matrix at once, we still find that naïve blockwise ternarization causes severe performance degradation. To investigate, we analyze the weight distributions and identify two key issues. **(i)** weights within a block often exhibit high variance, making ternarization too coarse and leading to large quantization error; **(ii)** many layers exhibit column-wise bias, where outlier columns distort the ternarization range and degrade fidelity.

**Structure-Aware Column Clustering.** To address these issues, we revisit *column reordering*, which in GPTQ (Frantar et al., 2023) is offered as an optional technique: while GPTQ can quantize weights in a fixed order, reordering columns by Hessian-derived importance has been shown to improve performance. Formally, reordering can be expressed using a permutation matrix $\mathbf{P}$:

$$\mathbf{W}' = \mathbf{W}\mathbf{P}, \quad \mathbf{X}' = \mathbf{X}\mathbf{P}, \quad \mathbf{X}'\mathbf{W}'^{\top} = \mathbf{X}\mathbf{W}^{\top}. \tag{14}$$

Here $\mathbf{P}$ simply permutes columns, so the result of matrix multiplication remains unchanged. Since applying $\mathbf{P}$ is just index reordering rather than actual multiplications, the computational overhead during inference is negligible. Building on this formulation, we observe that the potential of reordering remains underexplored. Placing structurally similar and numerically close columns in the same block yields a more compact distribution, improving row-wise ternarization. Similarly, grouping outliers together prevents them from distorting normal columns—outliers among outliers cease to be outliers. To this end, we propose a simple yet effective structure-aware column clustering method. Specifically, we compute pairwise cosine similarities between weight columns to capture their structural similarity:

$$S_{ij} = \frac{\mathbf{W}_{:,i}^{\top}\mathbf{W}_{:,j}}{\|\mathbf{W}_{:,i}\|_2 \|\mathbf{W}_{:,j}\|_2}, \tag{15}$$

where $\mathbf{W}_{:,i}$ denotes the $i$-th column of $\mathbf{W}$. Based on the similarity matrix $S$, we cluster columns with aligned directions to form more homogeneous blocks for ternarization. As shown in Fig. 3 (right), reordering reduces block variances, indicating more compact weight distributions within blocks.

**Efficient Integration with GPTQ.** GPTQ quantizes weights block by block, applying error compensation after each step. This inter-block dependency makes a one-time clustering-based reordering ineffective, while re-clustering after every update is too costly. To balance accuracy and efficiency, we adopt a lightweight strategy: after each update, we compute a mean reference from the residual and select the top-$k$ similar columns, where $k$ is the quantization block size:

$$\mathcal{B} = \text{Top-}k\left(\left\{\frac{\mathbf{W}_{:,i}^{\top}\bar{\mathbf{w}}}{\|\mathbf{W}_{:,i}\|_2 \|\bar{\mathbf{w}}\|_2}\right\}_{i=k}^{m}\right), \quad \text{with } \bar{\mathbf{w}} = \frac{1}{m}\sum_{i=k}^{m}\mathbf{W}_{:,i}, \tag{16}$$

Here, $\bar{\mathbf{w}}$ is the mean vector of the remaining submatrix, and $\mathcal{B}$ contains the top-$k$ most similar columns, forming the next quantization block. We term this lightweight strategy Structural Similarity-based Reordering (SSR), which retains the benefits of reordering with minimal overhead.

Table 1: **Evaluation on Multiple LLM Backbones.** We report perplexity (PPL) on WikiText2 and C4, and accuracy (%) on seven zero-shot tasks. All quantized models use a block size of 128. Best and second-best results (excluding FP16) are marked in **bold** and underlined, respectively.

| Model | Method | #W | Wiki2(↓) | C4(↓) | PiQA | Arc E | Arc C | Hella. | Wino. | OBQA | BoolQ | Avg(↑) |
|---|---|---|---|---|---|---|---|---|---|---|---|---|
| L-7B | FP16 | 16 | 5.68 | 7.34 | 78.67 | 75.34 | 41.89 | 56.93 | 70.01 | 34.20 | 75.05 | 61.73 |
| | AWQ | 2 | 2.60e5 | 2.86e5 | 52.83 | 25.25 | 22.44 | 25.27 | 49.88 | 14.00 | 37.83 | 32.50 |
| | GPTQ | 2 | 129.19 | 79.06 | 55.39 | 30.64 | 19.62 | 27.36 | 48.30 | 14.40 | 44.74 | 34.35 |
| | QuIP | 2 | 29.74 | 33.74 | N/A | N/A | N/A | N/A | N/A | N/A | N/A | N/A |
| | Slim-LLM | 2 | 14.58 | 30.71 | 57.83 | 33.46 | **25.09** | **36.70** | 52.64 | 16.40 | 56.05 | 39.74 |
| | PB-LLM | 1.7 | 82.76 | 76.63 | 55.17 | 29.42 | 18.94 | 27.68 | 47.83 | 12.20 | 42.87 | 33.44 |
| | PT²-LLM | 1.58 | **11.39** | **24.55** | **63.49** | **52.48** | 24.32 | 34.04 | **59.12** | **18.40** | **63.64** | **45.07** |
| L-13B | FP16 | 16 | 5.09 | 6.80 | 79.16 | 77.40 | 46.42 | 59.91 | 72.69 | 33.20 | 77.89 | 63.81 |
| | AWQ | 2 | 2.76e5 | 2.30e5 | 53.37 | 26.22 | 22.87 | 25.54 | 49.96 | 15.60 | 62.17 | 36.53 |
| | GPTQ | 2 | 20.46 | 18.97 | 61.97 | 41.67 | 24.06 | 38.13 | 54.85 | 19.20 | 47.13 | 41.00 |
| | QuIP | 2 | 38.82 | 28.62 | N/A | N/A | N/A | N/A | N/A | N/A | N/A | N/A |
| | Slim-LLM | 2 | 9.12 | 19.59 | 66.59 | 55.22 | 25.00 | 38.25 | 62.83 | 24.40 | 64.01 | 48.04 |
| | PB-LLM | 1.7 | 44.93 | 40.64 | 60.12 | 36.53 | 19.28 | 30.78 | 50.67 | 14.60 | 62.75 | 39.25 |
| | PT²-LLM | 1.58 | **9.11** | **17.32** | **67.19** | **58.50** | **26.19** | **39.17** | **63.54** | 22.20 | 63.67 | **48.64** |
| L-65B | FP16 | 16 | 3.53 | 5.81 | 81.28 | 81.36 | 52.82 | 64.55 | 77.43 | 38.20 | 84.83 | 68.64 |
| | AWQ | 2 | 7.40e4 | 7.50e4 | 53.16 | 25.13 | 22.27 | 25.48 | 49.80 | 11.20 | 37.83 | 32.12 |
| | GPTQ | 2 | 8.66 | 10.23 | 73.12 | 64.94 | 35.07 | 47.95 | 63.93 | 26.60 | 67.31 | 54.13 |
| | QuIP | 2 | 7.83 | 13.99 | N/A | N/A | N/A | N/A | N/A | N/A | N/A | N/A |
| | Slim-LLM | 2 | **6.15** | 11.11 | **75.20** | 53.72 | 35.10 | 45.91 | 70.21 | 25.80 | **76.91** | 54.69 |
| | PB-LLM | 1.7 | 12.81 | 15.30 | 72.18 | 68.43 | 35.18 | 46.35 | 70.16 | 26.40 | 62.75 | 54.49 |
| | PT²-LLM | 1.58 | 6.62 | **9.17** | 73.01 | 70.08 | **35.58** | 46.52 | 70.48 | **28.00** | 67.98 | **55.95** |
| L2-7B | FP16 | 16 | 5.47 | 7.26 | 78.07 | 76.30 | 43.34 | 57.16 | 68.98 | 31.40 | 77.68 | 61.85 |
| | AWQ | 2 | 2.22e5 | 1.70e5 | 52.39 | 25.00 | 21.16 | 25.58 | 49.09 | 18.00 | 37.83 | 32.72 |
| | GPTQ | 2 | 52.22 | 35.27 | 58.05 | 33.16 | 21.42 | 32.65 | 49.88 | 15.60 | 55.47 | 38.03 |
| | QuIP | 2 | 39.73 | 31.94 | N/A | N/A | N/A | N/A | N/A | N/A | N/A | N/A |
| | Slim-LLM | 2 | 15.84 | 84.92 | **63.82** | 47.81 | 23.38 | 33.76 | 56.91 | 17.80 | 59.97 | **43.35** |
| | PB-LLM | 1.7 | 66.41 | 80.69 | 53.59 | 27.82 | 18.69 | 26.91 | 48.54 | 13.20 | 41.25 | 32.86 |
| | PT²-LLM | 1.58 | **11.56** | **24.38** | 62.95 | 47.01 | 21.08 | 33.82 | 56.75 | 18.80 | 62.91 | 43.33 |
| L2-70B | FP16 | 16 | 3.32 | 5.71 | 82.15 | 82.79 | 54.44 | 64.78 | 77.98 | 37.20 | 83.76 | 69.01 |
| | AWQ | 2 | 7.25e4 | 7.30e4 | 52.50 | 25.76 | 22.35 | 25.33 | 49.49 | 14.20 | 62.17 | 35.97 |
| | GPTQ | 2 | 8.18 | 19.55 | 72.52 | 62.67 | 34.56 | **47.66** | 67.17 | 25.00 | **66.76** | 53.76 |
| | QuIP | 2 | N/A | N/A | N/A | N/A | N/A | N/A | N/A | N/A | N/A | N/A |
| | Slim-LLM | 2 | 6.28 | N/A | N/A | N/A | N/A | N/A | N/A | N/A | N/A | N/A |
| | PB-LLM | 1.7 | 28.37 | N/A | N/A | N/A | N/A | N/A | N/A | N/A | N/A | N/A |
| | PT²-LLM | 1.58 | **6.27** | **12.00** | **72.96** | **71.00** | **37.71** | 46.17 | **71.35** | **25.60** | 66.30 | **55.87** |
| L3-8B | FP16 | 16 | 6.14 | 9.45 | 79.54 | 80.13 | 50.34 | 60.13 | 73.40 | 34.60 | 81.01 | 65.59 |
| | AWQ | 2 | 1.70e5 | 2.10e5 | 52.72 | 24.16 | 21.50 | 25.58 | 49.33 | **14.60** | **62.17** | 35.72 |
| | GPTQ | 2 | 1480.43 | 394.74 | 52.12 | 25.72 | **21.59** | 26.72 | 49.17 | 13.60 | 44.16 | 33.30 |
| | QuIP | 2 | 84.97 | 130.00 | N/A | N/A | N/A | N/A | N/A | N/A | N/A | N/A |
| | Slim-LLM | 2 | 38.21 | 390.02 | 55.77 | 32.15 | 19.11 | 27.83 | 48.78 | 13.20 | 44.83 | 27.83 |
| | PB-LLM | 1.7 | 73.08 | **104.15** | 56.64 | 33.08 | 17.15 | 27.98 | 51.07 | 12.40 | 55.44 | 36.25 |
| | PT²-LLM | 1.58 | **32.19** | 129.83 | **56.86** | **34.22** | 18.43 | **30.36** | 53.28 | 13.80 | 57.58 | **37.79** |
| Qwen3 14B-Base | FP16 | 16 | 6.38 | 9.68 | 80.50 | 74.20 | 44.11 | 54.27 | 74.59 | 35.00 | 86.50 | 68.13 |
| | AWQ | 2 | 2.68e7 | 2.18e7 | 53.00 | 24.60 | 23.00 | 25.30 | 50.70 | 20.00 | 46.20 | 34.69 |
| | GPTQ | 2 | 37.90 | 74.50 | 56.31 | 34.64 | 20.65 | 33.30 | 52.72 | 17.20 | 46.33 | 37.31 |
| | QuIP | 2 | N/A | N/A | N/A | N/A | N/A | N/A | N/A | N/A | N/A | N/A |
| | Slim-LLM | 2 | 22.85 | 68.38 | 61.83 | 52.54 | **29.35** | 31.52 | 52.04 | 20.40 | 61.20 | 44.13 |
| | PB-LLM | 2 | 2.89e4 | 2.44e4 | 54.08 | 25.93 | 20.73 | 25.76 | 47.99 | 15.00 | 38.04 | 32.50 |
| | PT²-LLM | 1.58 | **16.48** | **68.13** | **62.95** | **53.03** | 23.63 | **33.65** | **59.75** | **20.60** | **62.17** | **45.11** |

# 4 EXPERIMENTS

## 4.1 EXPERIMENTAL SETTINGS

**Implementation Details.** All experiments are conducted using PyTorch (Paszke et al., 2019b) and Huggingface (Paszke et al., 2019a) on a single NVIDIA A800-80GB GPU. As PT²-LLM is a PTQ framework, it requires no training or gradient backpropagation. Following Li et al. (2025a) and Huang et al. (2025), we use 128 calibration samples from the Wikitext2 (Merity et al., 2017) dataset, each with a sequence length of 2048. All quantized models use a fixed block size of 128.

**Models and Evaluation.** We conduct comprehensive experiments on the LLaMA (Touvron et al., 2023a), LLaMA-2 (Touvron et al., 2023b), and LLaMA-3 families (Dubey et al., 2024), as well as the more recent Qwen3 series (Yang et al., 2025). Following prior work (Frantar et al., 2023; Lin et al., 2024b), we evaluate model performance in terms of both perplexity and accuracy. We report perplexity on WikiText2 (Merity et al., 2017) and C4 (Raffel et al., 2020) using a sequence length of 2048 tokens, and assess zero-shot accuracy on seven widely-used QA benchmarks: ARC-c (Clark et al., 2018), ARC-e (Clark et al., 2018), BoolQ (Clark et al., 2019), HellaSwag (Zellers et al., 2019), OBQA (Mihaylov et al., 2018), PIQA (Bisk et al., 2020), and Winogrande (Sakaguchi et al., 2020).

Table 2: Ablation studies conducted on LLaMA-2-7B and LLaMA-3-8B. We report perplexity on Wikitext2 and C4, as well as average accuracy across seven zero-shot tasks.

(a) Effectiveness of ITF and AGA

| Model | ITF | AGA | Wikitext2↓ | C4↓ | Avg. Acc↑ |
|---|---|---|---|---|---|
| LLaMA-2-7B | ✗ | ✗ | 22.88 | 222.17 | 37.11 |
| | ✓ | ✗ | 15.47 | 34.17 | 38.12 |
| | ✗ | ✓ | 12.25 | 26.17 | 42.86 |
| | ✓ | ✓ | 11.56 | 24.38 | 43.33 |
| LLaMA-3-8B | ✗ | ✗ | 247.75 | 1227.94 | 33.26 |
| | ✓ | ✗ | 83.76 | 1039.80 | 33.90 |
| | ✗ | ✓ | 47.83 | 520.14 | 35.29 |
| | ✓ | ✓ | 32.19 | 129.83 | 37.79 |

(b) Effectiveness of SSR

| Model | Reorder | Method | Wikitext2↓ | C4↓ | Avg. Acc↑ |
|---|---|---|---|---|---|
| LLaMA-2-7B | ✗ | - | 13.06 | 27.66 | 41.37 |
| | ✓ | Random | 12.84 | 28.24 | 40.86 |
| | ✓ | Hessian-based | 12.35 | 25.44 | 39.15 |
| | ✓ | SSR | 11.56 | 24.38 | 43.33 |
| LLaMA-3-8B | ✗ | - | 112.83 | 599.19 | 33.08 |
| | ✓ | Random | 113.42 | 466.07 | 33.37 |
| | ✓ | Hessian-based | 35.86 | 131.33 | 37.31 |
| | ✓ | SSR | 32.19 | 129.83 | 37.79 |

(c) Ablation study on calibration set size

| Model | Calib. Set Size | Wikitext2↓ | C4↓ | Avg. Acc↑ |
|---|---|---|---|---|
| LLaMA-2-7B | 64 | 11.92 | 25.27 | 43.31 |
| | 128 | 11.56 | 24.38 | 43.33 |
| | 256 | 11.35 | 23.52 | 43.55 |
| LLaMA-3-8B | 64 | 38.90 | 252.19 | 35.20 |
| | 128 | 32.19 | 129.83 | 37.79 |
| | 256 | 32.25 | 167.48 | 37.95 |

(d) Ablation study on calibration set type

| Model | Calib. Data Type | Wikitext2↓ | C4↓ | Avg. Acc↑ |
|---|---|---|---|---|
| LLaMA-2-7B | Wikitext2 | 11.56 | 24.38 | 43.33 |
| | C4 | 18.94 | 20.15 | 43.32 |
| | PTB | 27.52 | 35.15 | 41.15 |
| LLaMA-3-8B | Wikitext2 | 32.19 | 129.83 | 37.79 |
| | C4 | 168.81 | 72.00 | 37.80 |
| | PTB | 428.97 | 579.57 | 35.21 |

**Baselines.** We compare PT$^2$-LLM against a diverse set of representative PTQ methods operating in the 2-bit and sub-2-bit regimes. Slim-LLM (Huang et al., 2025) serves as a strong baseline for mixed-precision quantization, achieving high performance with an average of 2 bits. PB-LLM (Shang et al., 2024) targets the sub-2-bit regime and is closest to ours in average bitwidth, making it a relevant baseline. We further include GPTQ (Frantar et al., 2023) and AWQ (Lin et al., 2024b) as widely used baselines, along with QuIP (Chee et al., 2025), which targets 2-bit quantization.

## 4.2 COMPARISONS WITH STATE-OF-THE-ART METHODS

Table 1 summarizes the results of PT$^2$-LLM and baselines on LLaMA, LLaMA-2, LLaMA-3, and Qwen3-base reporting WikiText2/C4 perplexity, zero-shot accuracy on seven tasks (with average), and average bitwidth. Despite operating at the lowest bitwidth (1.58), PT$^2$-LLM consistently ranks among the top two in both perplexity and average accuracy across all model sizes. It clearly outperforms 2-bit baselines such as GPTQ, AWQ, and QuIP. Compared to Slim-LLM, the current SOTA 2-bit method, PT$^2$-LLM achieves higher average accuracy on all models except LLaMA-2-7B, where it performs comparably. Relative to PB-LLM with comparable bitwidth, PT$^2$-LLM delivers significant gains: on LLaMA-7B, it improves average accuracy from 33.44 to 45.07, reduces WikiText2 perplexity by 86%, and requires less memory. Additional results are provided in the supplementary file.

## 4.3 ABLATION STUDY

**Effectiveness of ITF and AGA.** We conduct a breakdown ablation to validate our Asymmetric Ternary Quantizer, focusing on its two components: Iterative Ternary Fitting (ITF) and Activation-aware Grid Alignment (AGA). As shown in Table 2a, ITF and AGA each provide gains over the baseline, for example, on LLaMA-2-7B, ITF reduces Wikitext2 perplexity from 22.88 to 15.47, while AGA improves average accuracy from 37.11 to 42.86. When applied together, they deliver the best overall performance, underscoring their complementary benefits and validating our design.

**Effectiveness of SSR.** We evaluate the impact of Structural Similarity-based Reordering (SSR) on quantization performance. As shown in Table 2b, omitting reordering yields suboptimal results, as ternarization is highly sensitive to scattered weights and outliers. Random reordering has minimal effect, while Hessian-based reordering, though occasionally effective, often overlooks the structural challenges of block-wise ternarization. In contrast, our proposed SSR consistently yields superior performance by promoting column-wise similarity, enabling tighter block-wise weight distributions and reducing outlier sensitivity in block-wise ternarization.

**Ablation for Calibration Set Size.** We assess the impact of calibration set size on PT$^2$-LLM's performance. As shown in Table 2c, larger sets yield modest gains in perplexity and accuracy. While 64 samples result in slightly lower performance, using 128 or 256 samples produces nearly identical results, indicating strong robustness to calibration size once a modest threshold is reached. Considering the trade-off between performance and memory efficiency, we adopt 128 samples for all experiments, demonstrating the practicality of PT$^2$-LLM in resource-constrained settings.

Table 3: Comparison of model size, reduction ratio, and compression time across various quantization methods on LLaMA-7B, highlighting the efficiency of PT$^2$-LLM over existing approaches.

| Method | FP16 | GPTQ | Slim-LLM | PB-LLM | BiLLM | ARB-LLM$_X$ | PT$^2$-LLM |
|---|---|---|---|---|---|---|---|
| Model Size | 13.48 GB | 2.19 GB | 2.30 GB | 2.91 GB | 2.93 GB | 3.23 GB | 1.88 GB |
| Size Reduction | - | 6.16× | 5.86× | 4.63× | 4.60× | 4.17× | 7.17× |
| Run Time | - | 21min | 182min | 22min | 45 min | 88 min | 32 min |

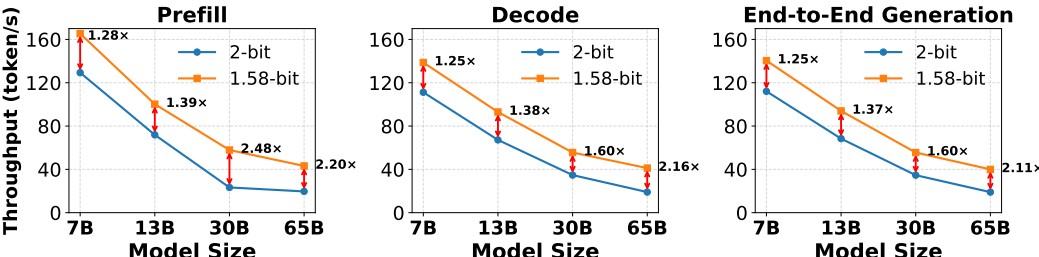

Figure 4: Throughput comparison between ternary (1.58-bit) and 2-bit quantized models across LLaMA 7B–65B. We evaluate performance on prefill, decode, and end-to-end generation stages.

**Ablation for Calibration Set Type.** We study how calibration dataset choice affects quantization by ablating over three datasets: WikiText2, C4, and PTB. As shown in Table 2d, using WikiText2 or C4 leads to comparable average accuracy, while PTB performs significantly worse, likely due to its lower data quality. Furthermore, calibrating on WikiText2 or C4 improves perplexity on their respective datasets, showing clear in-domain benefits. Considering all metrics and following prior work such as SliM-LLM (Huang et al., 2025), we adopt WikiText2 as our calibration set.

## 4.4 COMPRESSION TIME AND MODEL SIZE ANALYSES

We compare the compression time and model size of PT$^2$-LLM with GPTQ, Slim-LLM, PB-LLM, and two binarization methods, BiLLM (Huang et al., 2024) and ARB-LLM (Li et al., 2025b).

**Compression Time.** As shown in Table 3, PT$^2$-LLM offers a strong trade-off between quality and efficiency. Compressing LLaMA-7B takes only 32 minutes, which is significantly faster than Slim-LLM (182 min), BiLLM (45 min), and ARB-LLM$_X$ (88 min). While slightly slower than GPTQ and PB-LLM, the overhead remains modest and acceptable given the improved performance.

**Model Size.** As shown in Table 3, PT$^2$-LLM achieves the smallest model size (1.88 GB for LLaMA-7B), yielding a 7.17× reduction over FP16. It surpasses GPTQ (6.16×), Slim-LLM (5.86×), and PB-LLM (4.63×), and even outperforms binarization methods like BiLLM (4.60×) and ARB-LLM$_X$ (4.17×), whose compression ratios are limited by overhead from complex bitmap designs. A detailed discussion on memory composition and storage breakdown is provided in the supplementary file.

## 4.5 INFERENCE SPEED EVALUATION

As shown in Figure 4, we evaluate inference throughput on NVIDIA A800 GPUs across four model scales (LLaMA-7B to LLaMA-65B) using `llama.cpp`[1] , with sequence lengths of 128 (prefill), 256 (decode), and 128+256 (end-to-end). Compared to the standard 2-bit quantization, our 1.58-bit PT$^2$-LLM consistently improves throughput across all three stages and achieves up to 2.1× acceleration on LLaMA-65B for end-to-end generation. Experimental details are provided in the supplementary file.

## 5 CONCLUSION

In this paper, we propose PT$^2$-LLM, a post-training ternarization framework tailored for LLMs. Our method proposes an Asymmetric Ternary Quantizer (ATQ) with a two-stage pipeline, where Iterative Ternary Fitting (ITF) reduces quantization error in a training-free manner and Activation-aware Grid Alignment (AGA) aligns ternary outputs more closely with full-precision ones. We further propose Structural Similarity-based Reordering (SSR), a plug-and-play strategy that reduces quantization difficulty and alleviates the impact of outliers. Experimental results show that PT$^2$-LLM attains competitive accuracy compared to SOTA 2-bit PTQ methods while reducing model size and accelerating inference. This work establishes a strong baseline for ternarization in the PTQ setting of LLMs, pushing the boundary of sub-2-bit compression and laying the foundation for future research.

---

[1] https://github.com/ggml-org/llama.cpp

## ACKNOWLEDGMENTS

This work is supported by the National Natural Science Foundation of China (62501386, 625B1025, 625B1024) and also sponsored by CCF-Tencent Rhino-Bird Open Research Fund.

## ETHICS STATEMENT

The research conducted in the paper conforms, in every respect, with the ICLR Code of Ethics.

## REPRODUCIBILITY STATEMENT

We have provided implementation details in Section 4. We will also release all the code and models.

## LLM USAGE STATEMENT

Large Language Models (LLMs) were used solely for polishing writing. They did not contribute to the research content or scientific findings of this work.

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
