# Supplementary Material:
# PT²-LLM: Post-Training Ternarization for Large Language Models

**Xianglong Yan**[1][*], **Chengzhu Bao**[1][*], **Zhiteng Li**[1], **Tianao Zhang**[1], **Kaicheng Yang**[1],
**Haotong Qin**[2], **Ruobing Xie**[3], **Xingwu Sun**[3], **Yulun Zhang**[1][†]
[1]Shanghai Jiao Tong University, [2]ETH Zürich, [3]Tencent Hunyuan

## CONTENTS

---

[*]Equal contribution
[†]Corresponding authors: Yulun Zhang, yulun100@gmail.com

# A  FORMULATION AND DERIVATION OF THE ASYMMETRIC TERNARY QUANTIZER

## A.1  ASYMETRIC TERNARY INITIALIZATION

We define the ternarization objective as:

$$\alpha^*, \mathbf{T}^* = \arg\min_{\alpha, \mathbf{T}} \|\widetilde{\mathbf{W}} - \alpha\mathbf{T}\|_F^2, \tag{1}$$

where the centered weight matrix $\widetilde{\mathbf{W}}$ is computed by subtracting the row-wise mean:

$$\widetilde{\mathbf{W}} = \mathbf{W} - \mu, \quad \mu = \frac{1}{m}\sum_{j=1}^{m}\mathbf{W}_{:,j}. \tag{2}$$

Each ternary value $\mathbf{T}_{ij}$ is then determined using a fixed threshold $\Delta$:

$$\mathbf{T}_{ij} = \begin{cases} 1, & \text{if } \widetilde{\mathbf{W}}_{ij} > \Delta, \\ 0, & \text{if } |\widetilde{\mathbf{W}}_{ij}| \leq \Delta, \\ -1, & \text{if } \widetilde{\mathbf{W}}_{ij} < -\Delta. \end{cases} \tag{3}$$

Here, $\Delta$ serves as a positive threshold that determines which weights are set to $-1$, $0$, or $1$.

With this, the original problem reduces to optimizing over $\alpha$ and $\Delta$:

$$\alpha, \Delta = \arg\min_{\alpha \geq 0, \Delta \geq 0}\left(|\mathbf{I}_\Delta| \cdot \alpha^2 - 2\alpha \sum_{(i,j) \in \mathbf{I}_\Delta}|\widetilde{\mathbf{W}}_{ij}| + c_\Delta\right), \tag{4}$$

where $\mathbf{I}_\Delta = \{(i,j) \mid |\widetilde{\mathbf{W}}_{ij}| > \Delta\}$ is the index set of non-zero ternary entries, and $c_\Delta$ is a constant independent of $\alpha$.

The closed-form solution for $\alpha$ under fixed $\Delta$ is:

$$\alpha = \frac{1}{|\mathbf{I}_\Delta|}\sum_{(i,j) \in \mathbf{I}_\Delta}|\widetilde{\mathbf{W}}_{ij}|. \tag{5}$$

The optimal threshold $\Delta$ can be approximated by minimizing the squared error:

$$\Delta = \arg\min_{\Delta > 0}\left(\frac{1}{|\mathbf{I}_\Delta|}\left(\sum_{(i,j) \in \mathbf{I}_\Delta}|\widetilde{\mathbf{W}}_{ij}|\right)^2\right). \tag{6}$$

Since exact optimization is computationally expensive, we adopt a rule-of-thumb approximation:

$$\Delta \approx \frac{0.75}{m}\sum_{j=1}^{m}|\widetilde{\mathbf{W}}_{:,j}|. \tag{7}$$

## A.2  ITERATIVE TERNARY FITTING

To reduce the discrepancy between the ternarized weights $\mathbf{T}$ and the original full-precision weights $\mathbf{W}$, the following optimization problem is formulated:

$$\alpha, \mu = \arg\min_{\alpha,\mu} \|(\alpha\mathbf{T} + \mu - \mathbf{W})\|_F^2. \tag{8}$$

We rewrite the function as follows:

$$\|\alpha\mathbf{T} + \mu - \mathbf{W}\|_F^2 = \sum_{j}(\alpha_j\mathbf{t}_j + \mu_j - \mathbf{w}_j)^2 \tag{9}$$

$$= \sum_{j}(\mathcal{E}_w)_j \tag{10}$$

$$= \mathcal{E}_w, \tag{11}$$

where $\mathbf{t}_j$ denotes the $j$-th row of $\mathbf{T}$, and $\mathbf{w}_j$ is the corresponding row of $\mathbf{W}$.

Taking the gradient of the inner summand with respect to $\alpha_j$ and $\mu_j$ :

$$\frac{\partial \mathcal{E}_w}{\partial \alpha_j} = 2(\alpha_j \mathbf{t}_j + \mu_j \mathbf{1}^\top - \mathbf{w}_j)\mathbf{t}_j^\top, \tag{12}$$

$$\frac{\partial \mathcal{E}_w}{\partial \mu_j} = 2(\alpha_j \mathbf{t}_j + \mu_j \mathbf{1}^\top - \mathbf{w}_j)\mathbf{1}. \tag{13}$$

Since the optimal solution is obtained at the stationary point, we set $\frac{\partial \mathcal{E}_w}{\partial \alpha_j} = 0$ and $\frac{\partial \mathcal{E}_w}{\partial \mu_j} = 0$ :

$$\begin{bmatrix} \mathbf{t}_j \mathbf{t}_j^\top & \mathbf{1}^\top \mathbf{t}_j^\top \\ \mathbf{t}_j \mathbf{1} & \mathbf{1}^\top \mathbf{1} \end{bmatrix} \begin{bmatrix} \alpha_j \\ \mu_j \end{bmatrix} = \begin{bmatrix} \mathbf{w}_j \mathbf{t}_j^\top \\ \mathbf{w}_j \mathbf{1} \end{bmatrix}. \tag{14}$$

Summing over j, we can solve for the

$$\alpha = \frac{1}{ad - bc}(dy_1 - by_2), \quad \mu = \frac{1}{ad - bc}(ay_2 - cy_1), \tag{15}$$

where all operations are taken to be element-wise and

$$a = \mathbf{diag}(\mathbf{T}\mathbf{T}^\top), \quad b = c = \mathbf{T}\mathbf{1}, \quad d = \mathbf{1}^\top \mathbf{1}, \tag{16}$$

$$y_1 = \mathbf{diag}(\mathbf{W}\mathbf{T}^\top), \quad y_2 = \mathbf{diag}(\mathbf{W}\mathbf{1}), \tag{17}$$

By substituting $a, b, c, d, y_1, y_2$ into the above conditions, we obtain the closed-form solutions for $\alpha$ and $\mu$:

$$\alpha = \frac{\mathbf{1}^\top \mathbf{1} \cdot \mathbf{diag}(\mathbf{W}\mathbf{T}^\top) - (\mathbf{T}\mathbf{1}) \circ \mathbf{diag}(\mathbf{W}\mathbf{1})}{\mathbf{diag}(\mathbf{T}\mathbf{T}^\top) \cdot \mathbf{1}^\top \mathbf{1} - (\mathbf{T}\mathbf{1})^2}, \tag{18}$$

$$\mu = \frac{\left( \mathbf{diag}(\mathbf{T}\mathbf{T}^\top) \circ \mathbf{diag}(\mathbf{W}\mathbf{1}) - (\mathbf{T}\mathbf{1}) \circ \mathbf{diag}(\mathbf{W}\mathbf{T}^\top) \right)}{\mathbf{diag}(\mathbf{T}\mathbf{T}^\top) \cdot \mathbf{1}^\top \mathbf{1} - (\mathbf{T}\mathbf{1})^2}. \tag{19}$$

We define

$$\mathbf{Z} = \frac{\mathbf{W} - \mu}{\alpha}, \tag{20}$$

for every element in T, we update it as follows:

$$\mathbf{T}_{ij} = \underset{t \in \{-1,0,1\}}{arg\min} |\mathbf{Z}_{ij} - t|. \tag{21}$$

### A.3 ACTIVATION-AWARE GRID ALIGNMENT

To optimize the ternarized weights $\mathbf{T}$ based on the calibration data $\mathbf{X}$, we formulate the following problem:

$$\alpha, \mu = \underset{\alpha, \mu}{\arg\min} ||(\alpha \mathbf{T} + \mu - \mathbf{W})\mathbf{X}||_F^2, \tag{22}$$

We rewrite the problem as follows:

$$||(\alpha \mathbf{T} + \mu - \mathbf{W})\mathbf{X}||_F^2 = \sum_{b=1} \sum_{i} ||(\alpha \mathbf{T} + \mu - \mathbf{W})\mathbf{X}_{b,:,i}||_2^2 \tag{23}$$

$$= \sum_j \sum_b \sum_i \left[ (\alpha_j \mathbf{t}_j + \mu_j \mathbf{1}^\top - \mathbf{w}_j)\mathbf{X}_{b,:,i} \right]^2 \tag{24}$$

$$= \sum_j \sum_b \sum_i (\mathcal{E}_x)_{jbi} \tag{25}$$

$$= \mathcal{E}_x. \tag{26}$$

where $\mathbf{t}_j$ denotes the $j$-th row of $\mathbf{T}$, and $\mathbf{w}_j$ is the corresponding row of $\mathbf{W}$.

Taking the gradient of the inner summand with respect to $\alpha_j$ and $\mu_j$ :

$$\frac{\partial \mathcal{E}_x}{\partial \alpha_j} = 2[(\alpha_j \mathbf{t}_j + \mu_j \mathbf{1}^\top - \mathbf{w}_j)\mathbf{X}_{bi}]\mathbf{t}_j \mathbf{X}_{bi}, \tag{27}$$

$$\frac{\partial \mathcal{E}_x}{\partial \mu_j} = 2[(\alpha_j \mathbf{t}_j + \mu_j \mathbf{1}^\top - \mathbf{w}_j)\mathbf{X}_{bi}]\mathbf{1}^\top \mathbf{X}_{bi}. \tag{28}$$

To simplify the problem, we let $\mathbf{S}_j = \alpha_j \mathbf{t}_j + \mu_j \mathbf{1}^\top - \mathbf{w}_j$, then we derive

$$\frac{\partial \mathcal{E}_x}{\partial \alpha_j} = 2\sum_b \sum_i (\mathbf{S}_j \mathbf{X}_{bi})(\mathbf{t}_j \mathbf{X}_{bi}) \tag{29}$$

$$= 2\sum_b \sum_i \mathbf{S}_j \mathbf{X}_{bi} \mathbf{X}_{bi}^\top \mathbf{t}_j \tag{30}$$

$$= 2\mathbf{S}_j \sum_b \sum_i \mathbf{X}_{bi} \mathbf{X}_{bi}^\top \mathbf{t}_j \tag{31}$$

$$= 2\mathbf{S}_j \mathbf{C} \mathbf{t}_j, \quad \left( \mathbf{C} = \sum_b \sum_i \mathbf{X}_{bi} \mathbf{X}_{bi}^\top \right) \tag{32}$$

which leads to:

$$\frac{1}{2}\frac{\partial \mathcal{E}_x}{\partial \alpha_j} = (\mathbf{t}_j \mathbf{C} \mathbf{t}_j^\top)\alpha_j + (\mathbf{1}^\top \mathbf{C} \mathbf{t}_j^\top)\mu_j - \mathbf{w}_j \mathbf{C} \mathbf{t}_j, \tag{33}$$

$$\frac{1}{2}\frac{\partial \mathcal{E}_x}{\partial \mu_j} = (\mathbf{t}_j \mathbf{C} \mathbf{1})\alpha_j + (\mathbf{1}^\top \mathbf{C} \mathbf{1})\mu_j - \mathbf{w}_j \mathbf{C} \mathbf{1}. \tag{34}$$

Let $\frac{\partial \mathcal{E}_x}{\partial \alpha_j} = 0$ and $\frac{\partial \mathcal{E}_x}{\partial \mu_j} = 0$:

$$\begin{bmatrix} \mathbf{t}_j \mathbf{C} \mathbf{t}_j^\top & \mathbf{1}^\top \mathbf{C} \mathbf{t}_j^\top \\ \mathbf{t}_j \mathbf{C} \mathbf{1} & \mathbf{1}^\top \mathbf{C} \mathbf{1} \end{bmatrix} \begin{bmatrix} \alpha_j \\ \mu_j \end{bmatrix} = \begin{bmatrix} \mathbf{w}_j \mathbf{C} \mathbf{t}_j \\ \mathbf{w}_j \mathbf{C} \mathbf{1} \end{bmatrix}. \tag{35}$$

The result can be written as follows:

$$\begin{bmatrix} a_j & b_j \\ c_j & d_j \end{bmatrix} \begin{bmatrix} \alpha_j \\ \mu_j \end{bmatrix} = \begin{bmatrix} y_{1j} \\ y_{2j} \end{bmatrix}, \tag{36}$$

Summing over j:

$$\begin{bmatrix} a_1 & 0 & 0 & \cdots & b_1 & 0 & 0 & \cdots \\ 0 & a_2 & 0 & \cdots & 0 & b_2 & 0 & \cdots \\ \vdots & \vdots & \vdots & \vdots & \vdots & \vdots & \vdots & \vdots \\ 0 & 0 & 0 & a_n & 0 & 0 & 0 & b_n \\ c_1 & 0 & 0 & \cdots & d_1 & 0 & 0 & \cdots \\ 0 & c_2 & 0 & \cdots & 0 & d_2 & 0 & \cdots \\ \vdots & \vdots & \vdots & \vdots & \vdots & \vdots & \vdots & \vdots \\ 0 & 0 & 0 & c_n & 0 & 0 & 0 & d_n \end{bmatrix} \begin{bmatrix} \alpha \\ \mu \end{bmatrix} = \begin{bmatrix} y_1 \\ y_2 \end{bmatrix}, \tag{37}$$

from which the closed-form solutions are obtained:

$$\alpha = \frac{1}{ad - bc}(dy_1 - by_2), \quad \mu = \frac{1}{ad - bc}(ay_2 - cy_1), \tag{38}$$

where all operations are taken to be element-wise and

$$a = \mathbf{diag}(\mathbf{T}\mathbf{C}\mathbf{T}^\top), \quad b = c = \mathbf{T}\mathbf{C}\mathbf{1}, \quad d = \mathbf{1}^\top \mathbf{C}\mathbf{1}, \tag{39}$$

$$y_1 = \mathbf{diag}(\mathbf{W}\mathbf{C}\mathbf{T}^\top), \quad y_2 = \mathbf{W}\mathbf{C}\mathbf{1}. \tag{40}$$

We obtain the closed-form solutions:

$$\alpha = \frac{d \cdot (\mathbf{W} \circ \mathbf{T})\mathbf{S}\mathbf{1} - \mathbf{v} \circ (\mathbf{W}\mathbf{S}\mathbf{1})}{d \cdot \mathbf{T}^2\mathbf{S}\mathbf{1} - \mathbf{v}^2}, \quad \mu = \frac{\mathbf{T}^2\mathbf{S}\mathbf{1} \circ (\mathbf{W}\mathbf{S}\mathbf{1}) - \mathbf{v} \circ [(\mathbf{W} \circ \mathbf{T})\mathbf{S}\mathbf{1}]}{d \cdot \mathbf{T}^2\mathbf{S}\mathbf{1} - \mathbf{v}^2}, \tag{41}$$

where $d = \mathbf{1}^\top \mathbf{S}\mathbf{1}$ is a scalar, and $\mathbf{v} = \mathbf{T}\mathbf{S}\mathbf{1}$; $\mathbf{T}^2$ and $\mathbf{v}^2$ denote element-wise squares.

## A.4 OVERFITTING IN ACTIVATION-AWARE GRID ALIGNMENT

To improve the expressiveness and accuracy of ternarized models, we aim to minimize the activation-aware output error $\mathcal{E}_x$, which measures how well the quantized weights approximate the original full-precision weights when interacting with representative calibration data. Under this objective, we experiment with two strategies for updating the ternary code matrix $\mathbf{T}$.

**Gradient-based Projection.** The first approach treats $\mathbf{T}$ as a continuous variable during optimization. After computing the gradient of the loss with respect to $\mathbf{T}$, the result is projected back to the ternary set $-1, 0, 1$ through simple nearest rounding. While this method is conceptually straightforward, we observe that it often yields suboptimal solutions. The mismatch between continuous optimization and discrete rounding makes it difficult to consistently lower the reconstruction error across the entire matrix.

**Column-wise Greedy Update.** The second approach greedily updates one column of $\mathbf{T}$ at a time, keeping all other columns fixed. Each column-wise update becomes a low-dimensional subproblem with a closed-form optimal solution over ternary values. Empirically, this method substantially reduces the output more effectively than the projection method.

**Overfitting Phenomenon.** Surprisingly, we find that aggressive minimization of the reconstruction loss leads to overfitting. While the greedy strategy significantly lowers the error with respect to the calibration set, the final model performs worse on downstream tasks. This indicates that overly aligning ternary weights to calibration data may distort the model's inductive biases and impair generalization—especially in autoregressive or masked settings common in diffusion-based LLMs.

**Future Directions.** To address this overfitting, future work may explore regularization techniques that limit excessive fitting to calibration samples, curriculum-based updates that gradually refine $\mathbf{T}$, or multi-objective formulations that combine reconstruction loss with downstream task loss. Another promising direction is to precondition the $\mathbf{T}$ update using meta-learned or task-aware initializations.

**Table: Effect of T Optimization on Model Performance**

| T Update Method | Avg. Output Error $\mathcal{E}_x \downarrow$ | Avg. Accuracy $\uparrow$ |
|---|---|---|
| Without T Update | 128.3 | 0.4507 |
| Greedy Column-wise Update | **24.7** | 0.4213 |

Table 1: Evaluation on LLaMA-7B: While greedy column-wise updates to the ternary matrix $\mathbf{T}$ significantly reduce the activation-aware output error $\mathcal{E}_x$, they also lead to a notable drop in downstream average accuracy. This suggests a trade-off between low-level reconstruction and high-level generalization, highlighting the risk of overfitting under aggressive quantization.

## B   IMPLEMENTATION DETAILS FOR THROUGHPUT EXPERIMENTS

All throughput experiments are conducted on the `llama.cpp` platform using NVIDIA A800 GPUs. We leverage the platform's native support for ternary quantization operators as the basis for our extension. Specifically, we integrate our proposed *Asymmetric Ternary Quantizer* into the existing operator framework, ensuring compatibility with the platform's optimized low-bit inference path.

In addition to quantizer integration, we implement a runtime module that performs activation reordering during forward inference, in accordance with the *Structural Similarity-based Reordering (SSR)* strategy. This module dynamically adjusts the layout of activations to match the reordered column structure expected by SSR, enabling compatibility with ternary matrix operations. The entire logic is seamlessly integrated into the forward pass.

Our quantization pipeline targets all linear layers within the model, including projection and feedforward modules, while the activation values are preserved at 16-bit precision throughout inference. This design choice ensures a fair comparison with prior methods that retain higher-precision activations, and avoids confounding effects introduced by activation quantization. All modifications are implemented in C++ with CUDA acceleration preserved through the original backend.

## C   MEMORY FOOTPRINT ANALYSIS

To evaluate the storage efficiency of our ternarization scheme, we analyze the memory footprint introduced by each component in the quantized model.

**Ternary Weight Encoding.** We adopt a compact encoding strategy in which 5 ternarized values (each taking values from $\{-1, 0, +1\}$) are packed into a single 8-bit integer. This yields an effective average of **1.6 bits per weight element**.

**Per-block Quantization Parameters.** In addition to the ternary codes, each block of weights shares two 16-bit floating-point parameters: a scaling factor $\alpha$ and a shift $\mu$, which are used to reconstruct the approximate full-precision values. The block size is fixed (e.g., 128), and the overhead introduced by these scalars is amortized across all elements in the block, contributing only a small fraction to the overall storage.

**Unquantized Modules.** Following common practice in LLM compression, certain modules are kept in full precision to preserve model quality. In our implementation, both the `LayerNorm` layers and the `lm-head` (final projection layer) are left unquantized. These components constitute a small portion of the total parameters and do not significantly impact the compression ratio.

## D    DETAILED PROCEDURE OF THE ASYMMETRIC TERNARY QUANTIZER

**Algorithm 1** provides the complete pseudocode for the Asymmetric Ternary Quantizer (ATQ). It elaborates the full ternarization procedure, including initialization, iterative ternary fitting, and the activation-aware grid alignment (AGA) phase. This modular pipeline enables efficient and accurate weight ternarization while adapting to activation statistics from calibration data.

---

**Algorithm 1** Asymmetric Ternary Quantizer (ATQ)

---

Weight matrix $\mathbf{W} \in \mathbb{R}^{n \times m}$; calibration data $\mathbf{X} \in \mathbb{R}^{B \times L \times m}$ Quantized weight matrix $\widehat{\mathbf{W}} \in \mathbb{R}^{n \times m}$

**Step 1: Ternary Initialization**

Compute row-wise bias: $\mu \leftarrow \frac{1}{m} \sum_{j=1}^{m} \mathbf{W}_{:,j}$

Center weights: $\widetilde{\mathbf{W}} \leftarrow \mathbf{W} - \mu$

Compute threshold: $\Delta \leftarrow \frac{0.75}{m} \sum_{j=1}^{m} |\widetilde{\mathbf{W}}_{:,j}|$

Initialize ternary matrix: $\mathbf{T}_{ij} \leftarrow \begin{cases} 1 & \text{if } \widetilde{\mathbf{W}}_{ij} > \Delta \\ -1 & \text{if } \widetilde{\mathbf{W}}_{ij} < -\Delta \\ 0 & \text{otherwise} \end{cases}$

Initialize scale: $\alpha \leftarrow \frac{\sum_{j=1}^{m} \mathbf{T}_{:,j} \cdot \widetilde{\mathbf{W}}_{:,j}}{\sum_{j=1}^{m} |\mathbf{T}_{:,j}|}$

**Step 2: Iterative Ternary Fitting (ITF)**

$\mathbf{T}$ not converged Update grid parameters $(\alpha, \mu)$ using:

$$\alpha^* = \frac{m \cdot (\mathbf{W} \circ \mathbf{T})\mathbf{1} - (\mathbf{T1}) \circ (\mathbf{W1})}{m \cdot (\mathbf{T} \circ \mathbf{T})\mathbf{1} - (\mathbf{T1})^2}$$

$$\mu^* = \frac{(\mathbf{T} \circ \mathbf{T})\mathbf{1} \circ (\mathbf{W1}) - (\mathbf{T1}) \circ [(\mathbf{W} \circ \mathbf{T})\mathbf{1}]}{m \cdot (\mathbf{T} \circ \mathbf{T})\mathbf{1} - (\mathbf{T1})^2}$$

Update ternary matrix:

$$\mathbf{T}_{ij} \leftarrow \arg\min_{t \in \{-1, 0, 1\}} \left| \frac{\mathbf{W}_{ij} - \mu_i^*}{\alpha_i^*} - t \right|$$

**Step 3: Activation-aware Grid Alignment (AGA)**

Compute activation covariance: $\mathbf{S} \leftarrow \sum_{b,i} \mathbf{X}_{b,:,i} \mathbf{X}_{b,:,i}^\top$

Update $(\alpha, \mu)$ using:

$$\alpha^* = \frac{d \cdot (\mathbf{W} \circ \mathbf{T})\mathbf{S1} - \mathbf{v} \circ (\mathbf{WS1})}{d \cdot \mathbf{T}^2\mathbf{S1} - \mathbf{v}^2} \quad \mu^* = \frac{\mathbf{T}^2\mathbf{S1} \circ (\mathbf{WS1}) - \mathbf{v} \circ [(\mathbf{W} \circ \mathbf{T})\mathbf{S1}]}{d \cdot \mathbf{T}^2\mathbf{S1} - \mathbf{v}^2}$$

where $d = \mathbf{1}^\top \mathbf{S1}$, $\mathbf{v} = \mathbf{TS1}$

**Step 4: Final Quantization**

$\widehat{\mathbf{W}} \leftarrow \alpha\mathbf{T} + \mu$

$\widehat{\mathbf{W}}$

---

# E VISUALIZATION

## E.1 COLUMN-WISE DEVIATION PATTERNS

We visualize representative weight matrices from LLaMA-3-8B and observe clear column-wise deviation patterns 1. This column-wise asymmetry highlights the rationality of performing column-wise reordering prior to quantization, as it helps to better align the weight distribution and improve representational efficiency.

## E.2 ASYMMETRIC WEIGHT DISTRIBUTION VISUALIZATION

As shown in Figure 2, many weight matrices in LLaMA-3-8B exhibit clear asymmetric distributions. Rather than being centered around zero, some layers present skewed profiles where a significant portion of weights deviates toward one side. In particular, we observe long tails or outlier concentrations either on the positive or negative axis, highlighting the existence of asymmetry. This property suggests that symmetric quantization may be suboptimal, and motivates the need for asymmetric or distribution-aware quantization strategies.

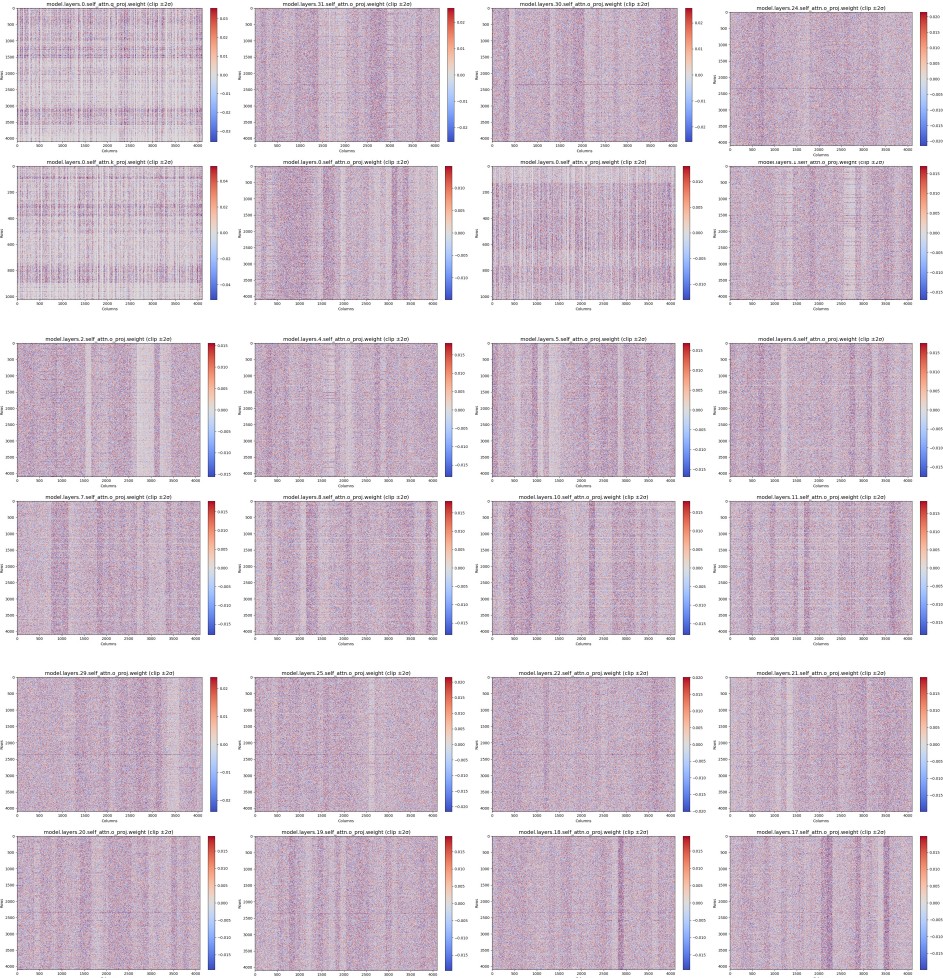

Figure 1: Visualization of column-wise deviation patterns across three representative layers.

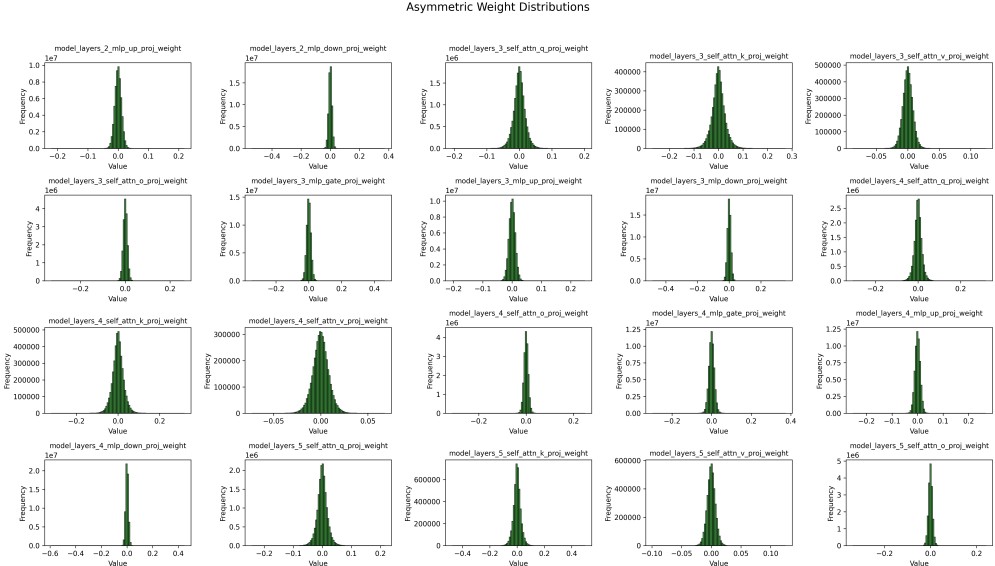

Figure 2: Histogram visualizations of selected weight matrices from LLaMA-3-8B.

## F  DIALOG EXAMPLES

**Prompt :** Shakespeare was a famous writer from ______

**Slim-LLM :** Shakespeare was a famous writer from the 19th century. He was a poet and a playwright. He was born in 1809 and died in 1849.

**GPTQ :** Shakespeare was a famous writer from a $100)

**PT$^2$-LLM:** Shakespeare was a famous writer from the 16th century. He wrote many plays and poems. Shakespeare is considered to be the greatest writer in the English language.

**Prompt :** Beethoven was a composer from ______

**Slim :** Beethoven was a composer from the 16th century. He was born in 1560 in Germany.

**GPTQ :** Beethoven was a composer from Homomal Homers

**PT:** Beethoven was a composer from the late 18th century. He was born in Bonn, Germany, and died in Vienna, Austria. Beethoven was a composer, pianist, and conductor. He was also a teacher.

**Prompt :** The Pyramids are located in ______

**Slim :** The Pyramids are located in the middle of the pyramid.

**GPTQ :** The Pyramids are located in the United States.

**PT$^2$-LLM:** The Pyramids are located in the Egyptian desert.

**Prompt :** The Great Barrier Reef is in ______

**Slim-LLM :** The Great Barrier Reef is in the Coral Sea, and is the largest single natural feature on earth.

**GPTQ :** The Great Barrier Reef is in the LATs.

**PT$^2$-LLM:** The Great Barrier Reef is in the north-east of Australia. It is the largest reef system in the world. It is 1,200 km long and 150 km wide.

**Prompt :** the chemical symbol for gold is______

**Slim-LLM :** The chemical symbol for gold is C. The chemical symbol for silver is Ag. The chemical symbol for copper is Cu. The chemical symbol for lead is Pb. The chemical symbol for mercury is Hg. The chemical symbol for platinum is Pt. The chemical symbol for tin is Sn. The chemical symbol for zinc is Zn. The chemical symbol for iron is Fe. The chemical symbol for nickel is Ni. The chemical symbol for chromium is Cr. The chemical symbol for bismuth is Bi.

**GPTQ :** the chemical symbol for gold is not the same as a person

**PT$^2$-LLM:** The chemical symbol for gold is Au. The symbol for silver is Ag. The symbol for iron is Fe. The symbol for lead is Pb. The symbol for zinc is Zn. The symbol for mercury is Hg. The symbol for copper is Cu. The symbol for nickel is Ni. The symbol for chromium is Cr.

**Prompt :** Michael Jordan played most of his career with the______

**Slim-LLM :** Michael Jordan played most of his career with the Washington Bullets. He was the first player to win the NBA Most Valuable Player Award.

**GPTQ :** Michael Jordan played most of his career with the The FBI

**PT$^2$-LLM:** Michael Jordan played most of his career with the Chicago Bulls, and was a member of the 1996 NBA All-Star team.

**Prompt :** The composer of "The Four Seasons" is ______

**Slim-LLM :** The Four Seasons is the composer of "The Four Seasons" is the composer of "The Four Seasons".

**GPTQ :** The composer of "The Four Seasons" is the "T" Night.

**PT$^2$-LLM:** The composer of "The Four Seasons" is Antonio Vivaldi, who was born in 1685. He was a composer, and he wrote "The Four Seasons" in 1725. He was born in Venice, Italy.

**Prompt :** Water boils at______

**Slim-LLM :** Water boils at the surface of the water.

**GPTQ :** Water boils at A

**PT$^2$-LLM:** Water boils at 100°C.

Table 2: **Evaluation on Multiple LLM Backbones.** We report perplexity (PPL) on WikiText2 and C4, and accuracy (%) on seven zero-shot tasks. All quantized models use a block size of 128.

| Model | Method | #W | Wiki2($\downarrow$) | C4($\downarrow$) | PiQA | Arc E | Arc C | Hella. | Wino. | OBQA | BoolQ | Avg($\uparrow$) |
|---|---|---|---|---|---|---|---|---|---|---|---|---|
| Qwen3-0.6B-base | FP16 | 16 | 12.70 | 17.10 | 70.00 | 65.60 | 33.90 | 41.10 | 58.50 | 24.60 | 69.70 | 51.91 |
| | AWQ | 2 | 6.42e7 | 9.85e7 | 53.20 | 24.80 | 21.90 | 25.70 | 50.80 | 5.00 | 46.30 | 32.53 |
| | GPTQ | 2 | 5.37e3 | 3.57e3 | 51.58 | 24.96 | 19.62 | 25.34 | 49.96 | 15.20 | 42.32 | 32.71 |
| | QuIP | 2 | N/A | N/A | N/A | N/A | N/A | N/A | N/A | N/A | N/A | N/A |
| | Slim-LLM | 2 | 282.12 | 1190.46 | 54.03 | 27.95 | 21.25 | 26.03 | 49.64 | 16.80 | 39.82 | 33.65 |
| | PB-LLM | 1.7 | 1.25e6 | 1.68e6 | 51.14 | 24.75 | 11.25 | 25.29 | 51.10 | 12.60 | 34.80 | 30.13 |
| | PT$^2$-LLM | 1.58 | 145.57 | 617.11 | 53.21 | 30.77 | 18.77 | 26.33 | 51.22 | 13.80 | 37.83 | 33.13 |
| Qwen3-1.7B-base | FP16 | 16 | 9.39 | 13.40 | 75.70 | 73.20 | 41.50 | 49.20 | 64.20 | 30.20 | 79.20 | 59.03 |
| | AWQ | 2 | 1.13e7 | 1.02e7 | 52.40 | 24.00 | 21.70 | 25.60 | 49.40 | 10.00 | 48.50 | 33.09 |
| | GPTQ | 2 | 101.69 | 375.74 | 52.12 | 27.23 | 18.94 | 26.29 | 52.96 | 17.80 | 44.98 | 34.33 |
| | QuIP | 2 | N/A | N/A | N/A | N/A | N/A | N/A | N/A | N/A | N/A | N/A |
| | Slim-LLM | 2 | 70.14 | 280.59 | 54.20 | 31.69 | 19.45 | 25.09 | 50.43 | 12.60 | 35.66 | 32.73 |
| | PB-LLM | 1.7 | 6.85e5 | 4.41e5 | 52.23 | 24.58 | 20.99 | 25.52 | 50.36 | 13.60 | 41.87 | 32.74 |
| | PT$^2$-LLM | 1.58 | 66.67 | 338.19 | 54.68 | 31.90 | 20.22 | 26.79 | 51.62 | 11.20 | 38.81 | 33.60 |
| Qwen3-4B-base | FP16 | 16 | 7.90 | 11.60 | 78.10 | 79.00 | 48.40 | 54.60 | 70.00 | 31.40 | 82.90 | 63.49 |
| | AWQ | 2 | 7.53e6 | 5.94e6 | 53.40 | 24.70 | 22.20 | 25.80 | 47.50 | 11.10 | 46.80 | 33.07 |
| | GPTQ | 2 | 65.17 | 193.61 | 52.61 | 27.53 | 20.73 | 27.38 | 51.46 | 17.00 | 45.50 | 34.60 |
| | QuIP | 2 | N/A | N/A | N/A | N/A | N/A | N/A | N/A | N/A | N/A | N/A |
| | Slim-LLM | 2 | 16.22 | 36.93 | 52.57 | 28.78 | 22.61 | 25.38 | 55.09 | 20.00 | 55.87 | 37.19 |
| | PB-LLM | 1.7 | 5.71e5 | 7.48e5 | 52.12 | 25.00 | 21.33 | 25.80 | 48.46 | 12.80 | 40.15 | 32.24 |
| | PT$^2$-LLM | 1.58 | 37.54 | 153.78 | 54.62 | 36.78 | 19.71 | 27.65 | 53.67 | 13.40 | 55.72 | 37.36 |
| Qwen3-8B-base | FP16 | 16 | 6.99 | 10.40 | 79.30 | 82.10 | 52.60 | 58.90 | 72.10 | 32.80 | 82.90 | 65.81 |
| | AWQ | 2 | 1.66e7 | 1.31e7 | 52.60 | 26.60 | 22.60 | 25.50 | 50.00 | 12.00 | 44.60 | 33.41 |
| | GPTQ | 2 | 106.41 | 347.02 | 57.40 | 30.60 | 20.50 | 33.40 | 52.50 | 16.40 | 48.70 | 37.07 |
| | QuIP | 2 | N/A | N/A | N/A | N/A | N/A | N/A | N/A | N/A | N/A | N/A |
| | Slim-LLM | 2 | 37.60 | 143.57 | 50.59 | 49.49 | 17.63 | 32.96 | 54.40 | 16.20 | 53.30 | 39.22 |
| | PB-LLM | 1.7 | 1.91e4 | 2.26e4 | 53.05 | 26.56 | 20.73 | 25.73 | 49.17 | 14.60 | 40.31 | 32.88 |
| | PT$^2$-LLM | 1.58 | 31.82 | 138.84 | 56.64 | 40.53 | 18.86 | 30.65 | 55.09 | 14.20 | 58.04 | 39.14 |
| L-30B | FP16 | 16 | 4.10 | 6.13 | 80.96 | 80.30 | 52.90 | 63.37 | 75.77 | 36.00 | 82.69 | 67.43 |
| | AWQ | 2 | 2.40e5 | 2.40e5 | 52.77 | 24.79 | 23.46 | 25.37 | 48.86 | 16.40 | 62.17 | 36.26 |
| | GPTQ | 2 | 15.29 | 14.86 | 62.84 | 42.80 | 22.01 | 39.74 | 53.59 | 19.40 | 54.80 | 42.17 |
| | QuIP | 2 | 22.30 | 19.33 | N/A | N/A | N/A | N/A | N/A | N/A | N/A | N/A |
| | Slim-LLM | 2 | 7.16 | 13.14 | 75.52 | 51.29 | 39.29 | 66.10 | 64.07 | 15.00 | 62.01 | 53.33 |
| | PB-LLM | 1.7 | 23.72 | 25.16 | 63.87 | 48.06 | 23.04 | 35.13 | 60.30 | 17.20 | 62.69 | 44.33 |
| | PT$^2$-LLM | 1.58 | 7.35 | 13.23 | 70.24 | 64.10 | 32.94 | 44.00 | 68.43 | 25.40 | 69.24 | 53.48 |
| L2-13B | FP16 | 16 | 4.88 | 6.73 | 79.05 | 79.42 | 48.38 | 60.03 | 72.22 | 35.20 | 80.55 | 64.98 |
| | AWQ | 2 | 1.20e5 | 9.40e4 | 53.26 | 26.09 | 22.70 | 25.57 | 48.93 | 13.40 | 62.17 | 36.02 |
| | GPTQ | 2 | 23.63 | 19.66 | 60.77 | 42.85 | 20.48 | 34.03 | 54.06 | 17.00 | 58.04 | 41.03 |
| | QuIP | 2 | 13.48 | 16.16 | N/A | N/A | N/A | N/A | N/A | N/A | N/A | N/A |
| | Slim-LLM | 2 | 9.98 | 22.62 | 66.54 | 52.09 | 26.37 | 33.83 | 50.06 | 21.40 | 64.74 | 45.00 |
| | PB-LLM | 1.7 | 236.42 | 184.67 | 54.68 | 27.36 | 18.86 | 26.58 | 50.43 | 13.20 | 38.20 | 32.76 |
| | PT$^2$-LLM | 1.58 | 9.93 | 21.49 | 63.76 | 54.38 | 24.23 | 34.23 | 58.01 | 18.80 | 62.23 | 45.09 |

# G  ADDITIONAL EXPERIMENTAL RESULTS

## G.1  RESULTS ON MORE LLMS

As shown in Table 2, we provide additional experimental results on other representative models. These experiments are intended to validate the generality and robustness of our method under varying model architectures and parameter scales. Unless otherwise specified, we follow the same quantization configurations and evaluation settings as in the main results.

## G.2  SENSITIVITY TO BLOCK SIZE

We conduct a sensitivity study on LLaMA-2-7B using block sizes $\{16, 32, 64, 128, 256\}$, evaluated on WikiText2 and C4 with sequence length $2048$. The results are summarized in Table 3. Smaller block sizes yield lower perplexity due to finer-grained approximation and reduced quantization error, at the cost of a modest increase in storage overhead. Unless otherwise specified, we adopt a block size of 128 as a practical trade-off between performance and efficiency, consistent with common settings in prior PTQ work.

## G.3  COMPARISON WITH TERNARYLLM WITHOUT TRAINING

Existing LLM ternarization approaches, including TernaryLLM Chen et al. (2024), do not provide open-source implementations. To ensure a fair comparison, we faithfully re-implement the *without-training* variant of TernaryLLM strictly following the algorithmic description in the original paper. We evaluate TernaryLLM (WO training) and our method on LLaMA-2-7B and LLaMA-3-8B, reporting perplexity on WikiText2 and C4 with sequence length $2048$. The results are summarized in Table 4.

Table 3: Sensitivity to block size on LLaMA-2-7B. Lower is better for perplexity.

| Model | Block Size | WikiText2 $\downarrow$ | C4 $\downarrow$ |
|---|---|---|---|
| LLaMA-2-7B | 16 | 8.07 | 13.75 |
| LLaMA-2-7B | 32 | 9.58 | 17.86 |
| LLaMA-2-7B | 64 | 10.84 | 21.82 |
| LLaMA-2-7B | 128 | 11.56 | 24.38 |
| LLaMA-2-7B | 256 | 11.90 | 25.59 |

Table 4: Comparison with TernaryLLM without training. Lower is better for perplexity.

| Model | Method | WikiText2 $\downarrow$ | C4 $\downarrow$ |
|---|---|---|---|
| LLaMA-2-7B | TernaryLLM (WO training) | 93.88 | 222.17 |
| LLaMA-2-7B | Ours | 11.56 | 24.38 |
| LLaMA-3-8B | TernaryLLM (WO training) | 247.75 | 1227.94 |
| LLaMA-3-8B | Ours | 32.19 | 129.83 |

The results indicate that the WO-training variant of TernaryLLM suffers severe degradation, whereas our method achieves substantially lower perplexity across both datasets and model scales.

## G.4 SENSITIVITY TO THE THRESHOLD

We provide a sensitivity analysis for the threshold used in Eq. (3) in the main paper, where the original setting uses a fixed coefficient $c$ (default $c = 0.75$). We treat $c$ as a tunable hyperparameter and vary it over a wide range. We additionally include a random initialization variant, where the threshold is sampled uniformly from a broad interval, to test robustness under intentionally uninformative initial conditions. Experiments are conducted on LLaMA-2-7B, reporting perplexity on WikiText2 and C4 with sequence length $2048$. Results are summarized in Table 5.

Table 5: Sensitivity to the threshold coefficient $c$ in Eq. (3) on LLaMA-2-7B. Lower is better for perplexity.

| Model | Threshold Coefficient $c$ | WikiText2 $\downarrow$ | C4 $\downarrow$ |
|---|---|---|---|
| LLaMA-2-7B | 0.1 | 11.59 | 24.54 |
| LLaMA-2-7B | 0.3 | 11.57 | 24.47 |
| LLaMA-2-7B | 0.5 | 11.56 | 24.45 |
| LLaMA-2-7B | 0.75 (default) | 11.56 | 24.38 |
| LLaMA-2-7B | 0.9 | 11.57 | 24.39 |
| LLaMA-2-7B | Random init. | 11.58 | 24.45 |

The results demonstrate that our method is robust to the choice of the threshold coefficient $c$: across the full range from $0.1$ to $0.9$, perplexity varies only marginally. Even with random initialization, the optimization reliably converges to a strong solution; poorer initialization may require more ITF iterations, but convergence remains stable, indicating that the ternary optimization does not rely on precise hyperparameter tuning.

REFERENCES

Tianqi Chen, Zhe Li, Weixiang Xu, Zeyu Zhu, Dong Li, Lu Tian, Emad Barsoum, Peisong Wang, and Jian Cheng. Ternaryllm: Ternarized large language model. *arXiv preprint arXiv:2406.07177*, 2024.