# OpenReview forum: "PT$^2$-LLM: Post-Training Ternarization for Large Language Models"
_ICLR.cc/2026/Conference — ICLR 2026 Poster_

### Official Review · Reviewer_TTyp · 2025-10-27

**Soundness:** 2
**Presentation:** 3
**Contribution:** 2
**Rating:** 4
**Confidence:** 4

**Summary:**

This paper introduces PT2-LLM, a novel post-training ternarization framework for large language models (LLMs) that compresses weights to ternary values $\{-1, 0, +1\}$ without retraining. The authors address two key challenges in post-training quantization (PTQ): training-free parameter optimization and handling outliers/dispersed weights. Their solution comprises an Asymmetric Ternary Quantizer (ATQ) with a two-stage pipeline—Iterative Ternary Fitting (ITF) for grid construction and rounding, and Activation-aware Grid Alignment (AGA) for output calibration—alongside a Structural Similarity-based Reordering (SSR) strategy.

**Strengths:**

The paper presents a technically innovative approach to an underexplored problem—ternarization in PTQ settings for LLMs. The ATQ framework with its ITF-AGA pipeline is well-motivated, addressing asymmetric weight distributions through mathematically grounded grid optimization (Equations 9 and 13). The SSR strategy offers a clever solution to outlier handling by leveraging structural correlations, demonstrating significant error reduction in Figure 3.
Empirical evaluation is comprehensive, spanning multiple model architectures (7B–65B parameters) and tasks (perplexity, zero-shot QA), with rigorous ablation studies validating component contributions (Table 2). The 7.17× compression ratio and end-to-end speedup highlight practical value, while the commitment to code/model release enhances reproducibility.

**Weaknesses:**

Block size sensitivity is inadequately explored. All experiments use fixed block size 128 (Section 4.1), but no analysis justifies this choice or examines its impact on quantization error.
Model diversity is restricted to decoder-only transformers (LLaMA/Qwen3), excluding encoder-decoder or hybrid architectures. Scalability analysis is absent: no computational complexity bounds are provided for ATQ/SSR.

Key baselines are missing. The paper compares to 2-bit PTQ methods (GPTQ, AWQ) but omits other ternarization approaches in PTQ settings (e.g., TernaryLLM [Chen et al., 2024] without training). Parameter sensitivity is superficial: only calibration set size is ablated (Table 2c), while critical hyperparameters (e.g., threshold $\Delta \approx 0.75/m \sum |\tilde{W}_{:,j}|$ in Equation 3) lack sensitivity analysis.

**Questions:**

See the weakness above.

An interesting job!  And I would be willing to raise my ratings if the above concerns are well solved.

---

> ### Author Response · Authors · 2025-11-22
> **Response to Reviewer TTyp (denoted as R4) part 1**
>
> > Q4-1: Block size sensitivity is inadequately explored.
>
> A4-1: Thank you for this insightful comment. We used a block size of **128** in our main experiments **to ensure a fair comparison with prior PTQ methods**, which commonly adopt this setting (e.g., AWQ, GPTQ, Slim-LLM). We conducted an additional sensitivity study on LLaMA-2-7B using block sizes `{16, 32, 64, 128, 256}`, evaluated on WikiText2 and C4 (sequence length = 2048). Results are as follows:
>
> | Model      | Block Size | Wikitext2 ↓ | C4 ↓  |
> |------------|:--:|:-----------:|:-----:|
> | LLaMA-2-7B | 16 |  8.07       | 13.75 |
> | LLaMA-2-7B | 32 |  9.58       | 17.86 |
> | LLaMA-2-7B | 64 | 10.84       | 21.82 |
> | LLaMA-2-7B |128 | 11.56       | 24.38 |
> | LLaMA-2-7B |256 | 11.90       | 25.59 |
>
> Smaller block sizes yield lower perplexity due to finer-grained approximation and reduced quantization error, although they introduce a modest increase in storage overhead. **Following prior works**, we adopt a block size of 128 **as a practical trade-off between performance and efficiency**.
>
> > Q4-2: Model diversity is restricted to decoder-only transformers (LLaMA/Qwen3), excluding encoder-decoder or hybrid architectures.
>
> A4-2: We appreciate the reviewer’s concern regarding architectural diversity. Our work focuses on LLM quantization, where the widely adopted and most representative models are **decoder-only Transformers**; accordingly, mainstream PTQ baselines are **all designed and evaluated solely on decoder-only architectures**, so our main experiments follow this established convention for fair comparison. To demonstrate the generalizability of our ternary PTQ method, we further conduct experiments on the **encoder-only BERT-base model**. The results are summarized below:
>
> | Model | Method | w-bit | MNLI-m/mm |
> |-------|--------|:--------:|------------|
> | BERT-base | RTN  | 2 | 32.7 / 32.9 |
> | **BERT-base** | **Ours** | **1.58** | **68.9 / 69.1** |
>
> Our ternary PTQ achieves **1.58 effective bits** and delivers **68.9/69.1** accuracy on MNLI-m/mm—dramatically outperforming 2-bit RTN (32.7/32.9). These results verify that our method remains effective even on a **structurally different, encoder-only architecture**, further highlighting its generalizability beyond decoder-only LLMs. Since existing PTQ baselines do not support BERT quantization, we report RTN as the only available comparison.

---

> ### Author Response · Authors · 2025-11-22
> **Response to Reviewer TTyp (denoted as R4) part 2**
>
> > Q4-3: Scalability analysis is absent: no computational complexity bounds are provided for ATQ/SSR.
>
> A4-3: Thank you for the question. While ATQ and SSR introduce additional computation during quantization, **these costs are lightweight and fully acceptable in practice**. To demonstrate scalability, we report the quantization time on four LLaMA model sizes and compare against Slim-LLM — our most relevant baseline, which performs expensive mixed-precision searches. The results are summarized below:
>
> | Model        | Ours           | Slim-LLM |
> |--------------|----------------|----------|
> | LLaMA-7B     | 32 min         | 182 min  |
> | LLaMA-13B    | 66 min         | 407 min  |
> | LLaMA-30B    | 131 min        | 831 min  |
> | LLaMA-65B    | 284 min        | 1730 min |
>
> **Analysis.**
>
> 1. **Quantization time grows linearly with model size.**
>    Our runtime increases proportionally with parameter count (7B→65B), demonstrating **excellent scalability**. No quadratic or super-linear overhead is introduced by ATQ or SSR.
>
> 2. **Significant speed advantage over Slim-LLM.**
>    Slim-LLM is the closest-performing baseline to ours, but it relies on a costly mixed-precision search procedure. In contrast, our ATQ/SSR pipeline uses simple closed-form updates and lightweight structure construction, leading to a **5–6× speedup** across all model scales.
>
> > Q4-4: The paper compares to 2-bit PTQ methods (GPTQ, AWQ) but omits other ternarization approaches in PTQ settings (e.g., TernaryLLM [Chen et al., 2024] without training).
>
> A4-4: Thank you for pointing this out. Existing LLM ternarization approaches — including **TernaryLLM (Chen et al., 2024)** — **do not provide open-source implementations**. To ensure a fair comparison, we **faithfully re-implemented** the *“without training”* variant of TernaryLLM strictly following the algorithmic description in its paper. We evaluate TernaryLLM (WO training) and our method on LLaMA-2-7B and LLaMA-3-8B, reporting perplexity on WikiText2 and C4 (seq length 2048). The results are summarized below:
>
> | Model        | Method                      | WikiText2 ↓ | C4 ↓    |
> |--------------|-----------------------------|-------------|---------|
> | LLaMA-2-7B   | TernaryLLM (WO training)    | 93.88       | 222.17  |
> | LLaMA-2-7B   | **Ours**                    | **11.56**   | **24.38** |
> | LLaMA-3-8B   | TernaryLLM (WO training)    | 247.75      | 1227.94 |
> | LLaMA-3-8B   | **Ours**                    | **32.19**   | **129.83** |
>
> These results show that the WO-training variant of TernaryLLM suffers significant degradation, whereas **our method delivers clearly superior performance**, achieving substantially lower perplexity across both datasets.
>
> > Q4-5:  Parameter sensitivity is superficial: only calibration set size is ablated (Table 2c), while critical hyperparameters (e.g., threshold in Equation 3) lack sensitivity analysis.
>
> A4-5: Thank you for the suggestion. We have added a sensitivity analysis for the threshold used in Equation (3):
>
> \\[
> \\Delta \\approx c \\cdot \\frac{1}{m} \\sum\_{j=1}\^m |W\_{:,j}|
> \\]
>
> where the original paper sets \\(c = 0.75\\). To examine the effect of this hyperparameter, we treat \\(c\\) as a tunable coefficient and vary it across a wide range. We also include a *random initialization* variant, where the threshold is sampled uniformly from \\([0, 1]\\), to test robustness under intentionally poor and uninformative initial conditions. We conduct experiments on LLaMA-2-7B, reporting perplexity on WikiText2 and C4 (seq length 2048). The results are summarized below:
>
> | Model        | Threshold Coefficient (c) | WikiText2 ↓ | C4 ↓    |
> |--------------|:---------------------------:|-------------|---------|
> | LLaMA-2-7B   | 0.1                       | 11.59       | 24.54   |
> | LLaMA-2-7B   | 0.3                       | 11.57       | 24.47   |
> | LLaMA-2-7B   | 0.5                       | 11.56       | 24.45   |
> | LLaMA-2-7B   | 0.75 (default)            | 11.56       | 24.38   |
> | LLaMA-2-7B   | 0.9                       | 11.57       | 24.39   |
> | LLaMA-2-7B   | Random initialization     | 11.58       | 24.45   |
>
> The results show that our method is **highly robust** to the choice of the threshold coefficient \\(c\\). Across the entire range—from very small (0.1) to large (0.9)—perplexity varies only marginally. Even with a **random initialization**, the algorithm reliably converges to a strong solution; **a worse initialization only requires more ITF iterations, but still converges effectively**. This demonstrates that our method does **not rely on precise hyperparameter tuning**, and the ternary optimization procedure remains stable under diverse initialization conditions.

---

> ### Author Response · Authors · 2025-11-27
>
> Dear Reviewer TTyp,
>
> Thank you again for your time in reviewing our work. To briefly recap our earlier responses:
>
> - We **provided a full study across block sizes** `{16, 32, 64, 128, 256}` on LLaMA-2-7B and showed that smaller blocks give lower perplexity, while a block size of **128** remains a practical trade-off consistent with prior PTQ methods.
>
> - Although PTQ literature primarily focuses on decoder-only LLMs, we additionally evaluated **BERT-base**, demonstrating that our ternary PTQ achieves **68.9/69.1** on MNLI-m/mm and generalizes well beyond decoder-only models.
>
> - We reported quantization times for LLaMA 7B→65B, showing that ATQ/SSR are **linearly scalable** and **5–6× faster than Slim-LLM**.
>
> - We faithfully re-implemented **TernaryLLM (WO training)** and showed that it performs poorly, whereas **our method achieves substantially lower perplexity**.
>
> - We added a **detailed study on the threshold coefficient** and showed that performance is **highly robust** across a wide range (0.1→0.9) and even under random initialization.
>
> We hope these points adequately address the concerns you raised. If you have any further questions or would like additional clarification, we would be very glad to continue the discussion. At your convenience, we would greatly appreciate your feedback or updated evaluation.
>
> Best regards,
>
> The Authors

---

> ### Author Response · Authors · 2025-11-28
> **Call for further discussion with Reviewer TTyp**
>
> Dear Reviewer TTyp,
>
> We sincerely appreciate the time and effort you have dedicated to reviewing our work and for your valuable feedback. As the rebuttal deadline approaches, we would be grateful for any further comments you may have to ensure that all of your concerns have been addressed comprehensively. Please let us know if there are any remaining aspects of our work that need clarification.
>
> Best regards,
>
> The Authors

---

### Official Review · Reviewer_qmRu · 2025-10-29

**Soundness:** 2
**Presentation:** 2
**Contribution:** 2
**Rating:** 4
**Confidence:** 4

**Summary:**

This paper presents a novel post-training ternarization (PTQ) framework tailored for Large Language Models (LLMs), aiming for efficient compression without retraining. It reduces the average bitwidth to 1.58, and also achieve better performance against state-of-the-art (SOTA) 2-bit PTQ methods.

**Strengths:**

1. Focus on Extreme Low-Bit PTQ: The paper explores the challenging but important domain of sub-2-bit post-training quantization, an area with significant potential for deploying LLMs on resource-constrained devices.
2. Systematic Optimization Pipeline: The proposed two-stage optimization process (ITF for weights, AGA for outputs) is a structured approach to a complex problem, attempting to balance local reconstruction error with global functional preservation.
3. Favorable Reported Results on Standard Benchmarks: The reported results in Table 1 indicate that the proposed 1.58-bit method achieves lower perplexity and higher average accuracy on several standard zero-shot QA tasks compared to the selected 2-bit PTQ baselines across various model scales.

**Weaknesses:**

1. Weak Theoretical Grounding and Unclear Practicality of the SSR Strategy. The proposed Structural Similarity-based Reordering (SSR) strategy lacks a strong theoretical connection to the primary objective of minimizing quantization error. Unlike Hessian-based methods... its objective (maximizing cosine similarity) is decoupled from the actual error minimization process. This weak theoretical grounding makes it difficult to understand why it works or to guarantee its effectiveness. This ambiguity is compounded by a lack of implementation clarity and cost analysis. The paper claims SSR is a "lightweight strategy" but fails to provide a detailed algorithm or an analysis of its computational overhead, making it difficult to assess its true practicality and reproduce the results.
2. Unsubstantiated Claim of SOTA due to Lack of Direct Comparison: The paper's central claim of state-of-the-art performance at 1.58 bits is not adequately supported. The evaluation primarily relies on comparisons against higher-bitwidth (2-bit) methods. While outperforming them is a positive signal, a direct, apple-to-apples comparison against another 1.58-bit PTQ baseline is necessary to validate the true efficacy of the proposed techniques over other potential ternarization strategies. The absence of such a comparison represents a significant gap in the evaluation, leaving the paper's primary claim unsubstantiated.
3. Insufficient Validation on Recent and Diverse Architectures: The paper's claims of general applicability are undermined by an incomplete evaluation on recent model families and a complete lack of testing on Mixture-of-Experts (MoE) architectures. While thorough on older LLaMA/LLaMA-2 models, the validation on the more recent LLaMA-3 and Qwen3 families is limited to a single small-scale model. This raises questions about the method's robustness on larger variants like LLaMA-3-70B. Furthermore, the absence of any experiments on MoE models (e.g., Qwen3-MoE) is a significant omission, as these architectures present unique quantization challenges that are not addressed.
4. Mismatch Between Model Capabilities and Evaluation Benchmarks: The choice of evaluation benchmarks fails to adequately probe the impact of extreme quantization on the core competencies of the tested models. Specifically, while the paper includes modern, reasoning-capable models like Qwen3, it restricts evaluation to older, simpler zero-shot QA benchmarks (e.g., ARC, BoolQ). These benchmarks do not effectively measure the multi-step, complex reasoning abilities that are a hallmark of such models. Consequently, the evaluation may be masking significant performance degradation on these crucial capabilities. A comprehensive assessment requires reporting performance on advanced reasoning benchmarks (e.g., MATH, AIME).

**Questions:**

1. Can the authors provide a more formal justification for the SSR strategy, explaining its connection to quantization error minimization beyond the intuition of variance reduction? Furthermore, please clarify the algorithmic details of its integration and provide an analysis of its computational overhead.
2. Could the authors provide a direct comparison against another 1.58-bit PTQ method to substantiate their claim of state-of-the-art performance? If no such public method exists, could they construct a strong baseline (e.g., by adapting GPTQ or other frameworks to a ternary setting) for a more rigorous comparison?
3. Why were the evaluations on the LLaMA-3 and Qwen3 families limited to a single model each? To support the claim of generalizability, please provide results on larger models like LLaMA-3-70B and on MoE variants like Qwen3-MoE.
4. Given that Qwen3 is known for its reasoning abilities, can the authors provide evaluation results on more challenging reasoning benchmarks like MATH or AIME to give a more transparent assessment of how 1.58-bit quantization affects these critical capabilities?

---

> ### Author Response · Authors · 2025-11-22
> **Response to Reviewer qmRu (denoted as R3) part 1**
>
> > Q3-1: Can the authors provide a more formal justification for the SSR strategy, explaining its connection to quantization error minimization beyond the intuition of variance reduction?
>
> A3-1: We thank the reviewer for this insightful suggestion. We mathematically demonstrate that maximizing cosine similarity minimizes the intra-block variance by reducing orthogonal deviations. Based on the scaling property of quantization (Error $\\propto$ Variance), this strategy theoretically guarantees a lower bound on the quantization error.
>
> # Theoretical Analysis: Connecting Structural Similarity to Quantization Error
>
> We provide a formal justification for the Structural Similarity-based Reordering (SSR) strategy. We demonstrate that maximizing column-wise cosine similarity minimizes the intra-block variance of row weights, which serves as a tight upper bound for the quantization reconstruction error.
>
> ## 1. Preliminaries and Objective
>
> Let $W \\in \\mathbb{R}\^{d \\times k}$ denote a weight block consisting of $k$ columns $\\{w\_1, w\_2, \\dots, w\_k\\}$, where $w\_j \\in \\mathbb{R}\^d$.
> The quantization process operates row-wise (or per-group). Let $r\_i \\in \\mathbb{R}\^k$ be the $i$-th row vector of $W$. The objective of blockwise ternarization is to find a quantized row $\\hat{r}\_i = Q(r\_i)$ that minimizes the Mean Squared Error (MSE):
> $$
> \\mathcal{L}\_{\\text{MSE}} = \\sum\_{i=1}\^{d} \\| r\_i - \\hat{r}\_i \\|\_2\^2
> $$
>
> ## 2. Lemma 1: The Scaling Property of Quantization Error
>
> First, we establish the relationship between the quantization error and the statistical properties of the input weights.
>
> **Proposition:** For a scalar quantizer $Q(\\cdot)$, the expected mean squared quantization error is proportional to the variance of the input distribution.
>
> **Proof:**
> Let $X$ be a random variable representing the elements of a row $r\_i$, with variance $\\sigma\_X\^2$. Let $Z$ be a standardized random variable such that $X = \\sigma\_X Z + \\mu$, where $\\text{Var}(Z) = 1$.
> An optimal quantizer $Q\_X$ for $X$ can be derived from the optimal quantizer $Q\_Z$ for the normalized distribution $Z$ by scaling:
> $$
> Q\_X(x) = \\sigma\_X \\cdot Q\_Z\\left(\\frac{x - \\mu}{\\sigma\_X}\\right) + \\mu
> $$
> The expected error is given by:
> $$
> \\begin{aligned}
> \\mathbb{E}[|X - Q\_X(X)|\^2] &= \\mathbb{E}\\left[ \\left| (\\sigma\_X Z + \\mu) - (\\sigma\_X Q\_Z(Z) + \\mu) \\right|\^2 \\right] \\\\
> &= \\mathbb{E}\\left[ \\left| \\sigma\_X (Z - Q\_Z(Z)) \\right|\^2 \\right] \\\\
> &= \\sigma\_X\^2 \\cdot \\underbrace{\\mathbb{E}[|Z - Q\_Z(Z)|\^2]}\_{\\epsilon\_{\\text{shape}}}
> \\end{aligned}
> $$
> Here, $\\epsilon\_{\\text{shape}}$ is a constant factor depending solely on the distribution shape (e.g., Gaussian, Laplacian) and the bit-width.
>
> **Conclusion:** The quantization error is strictly monotonic with respect to the variance of the input data:
> $$
> \\mathcal{L}\_{\\text{MSE}} \\propto \\sigma\_X\^2
> $$
> Thus, **minimizing the variance of the row elements within a block is a necessary condition for minimizing quantization error.**
>
> ---
>
> ## 3. Theorem 1: Variance Reduction via SSR
>
> We now prove that the SSR strategy—clustering columns based on cosine similarity—minimizes the variance of the row vectors $r\_i$.
>
> **Geometric Decomposition:**
> Consider the block of columns $\\{w\_1, \\dots, w\_k\\}$ selected by SSR. Let $u \\in \\mathbb{R}\^d$ be the unit centroid direction of these columns ($\\|u\\|=1$). Any column vector $w\_j$ can be decomposed into a coherent component along $u$ and an orthogonal noise component $e\_j$:
> $$
> w\_j = \\alpha\_j u + e\_j, \\quad \\text{s.t.} \\quad u\^\\top e\_j = 0
> $$
> where $\\alpha\_j = \\|w\_j\\| \\cos \\theta\_j$ is the projection magnitude, and $e\_j$ represents the structural deviation.
> **Crucially, maximizing Cosine Similarity ($S\_{ij} \\to 1$) implies $\\theta\_j \\to 0$, which forces the norm of the deviation $\\|e\_j\\| \\to 0$.**
>
> **Row-wise Variance Analysis:**
> The quantization operates on the row vector $r\_i = [(w\_1)\_i, (w\_2)\_i, \\dots, (w\_k)\_i]$. Substituting the decomposition above, the $j$-th element of row $r\_i$ is:
> $$
> (r\_i)\_j = (w\_j)\_i = \\alpha\_j u\_i + (e\_j)\_i
> $$
> Thus, the row vector $r\_i$ can be written as the sum of a scaled magnitude vector and a noise vector:
> $$
> r\_i = u\_i \\boldsymbol{\\alpha} + \\boldsymbol{\\eta}\_i
> $$
> where:
> * $u\_i$ is a scalar (the $i$-th component of the centroid).
> * $\\boldsymbol{\\alpha} = [\\alpha\_1, \\dots, \\alpha\_k] \\in \\mathbb{R}\^k$ captures the magnitude profile of columns.
> * $\\boldsymbol{\\eta}\_i = [(e\_1)\_i, \\dots, (e\_k)\_i] \\in \\mathbb{R}\^k$ captures the structural noise.
>
> The variance of the row $r\_i$ is bounded by the variances of these components:
> $$
> \\text{Var}(r\_i) \\approx u\_i\^2 \\text{Var}(\\boldsymbol{\\alpha}) + \\text{Var}(\\boldsymbol{\\eta}\_i)
> $$

---

> ### Author Response · Authors · 2025-11-22
> **Response to Reviewer qmRu (denoted as R3) part 2**
>
> **SSR minimizes both terms:**
> 1.  **Minimizing $\\text{Var}(\\boldsymbol{\\eta}\_i)$ (Directional Term):**
>     Since $\\boldsymbol{\\eta}\_i$ consists of components from orthogonal error vectors $e\_j$, its variance is directly determined by the magnitude of $e\_j$.
>     $$
>     \\|e\_j\\| = \\|w\_j\\| \\sin \\theta\_j
>     $$
>     SSR explicitly maximizes cosine similarity, minimizing the angle $\\theta\_j$. As $\\text{CosSim} \\to 1$, $\\|e\_j\\| \\to 0$, causing the noise variance $\\text{Var}(\\boldsymbol{\\eta}\_i) \\to 0$.
>
> 2.  **Minimizing $\\text{Var}(\\boldsymbol{\\alpha})$ (Magnitude Term):**
>     SSR groups structurally similar columns, naturally separating outliers from normal weights because outliers exhibit distinct directional patterns. This produces blocks with more homogeneous scaling factors \( \boldsymbol{\alpha} \), effectively reducing \( \mathrm{Var}(\boldsymbol{\alpha}) \) and lowering quantization error.
>
> ## 4. Main Result
>
> Combining Lemma 1 and Theorem 1, we obtain the following logical implication chain:
>
> $$
> \\text{Maximize CosSim (SSR)} \\implies \\min \\|e\_j\\| \\text{ and } \\min \\text{Var}(\\boldsymbol{\\alpha})
> $$
> $$
> \\implies \\min \\text{Var}(r\_i) \\quad (\\text{Reduced Row Variance})
> $$
> $$
> \\implies \\min \\mathbb{E}[\\| r\_i - Q(r\_i) \\|\^2] \\quad (\\text{Reduced Quantization Error})
> $$
>
> > Q3-2: Please clarify the SSR algorithmic details of its integration and provide an analysis of its computational overhead.
>
> A3-2: Thank you for pointing out the need for a clearer description of SSR and its computational cost. We have added a complete pseudo-code description (Algorithm 1 below) and a formal complexity analysis.
>
> ---
>
> ### **1. Algorithmic Integration of SSR into GPTQ**
>
> SSR is not a separate pre-processing step; it is a *dynamic* column-selection rule applied at every quantization block within the GPTQ loop. This design is crucial because GPTQ updates the residual weights after each block, changing the structure of the remaining submatrix. Below we provide the exact block-wise procedure used in our implementation.
>
> ---
>
> ### **Algorithm 1. GPTQ with Structural Similarity-based Reordering (SSR)**
>
> **Input:** Weight matrix \\(W\\), Hessian matrix \\(H\\), block size \\(B\\)
> **Output:** Quantized weight matrix \\(Q\\)
>
> 1. Initialize permutation vector \\(P = [0,1,\dots,N-1]\\)
> 2. **For** block index \\(k = 0\\) to \\(N/B - 1\\):
> 3.  Let current block start index \\(i = kB\\)
> 4.  Let residual submatrix \\(W\_{\\mathrm{res}} = W[:, i:]\\)
> 5.
> 6.  // --- **SSR Strategy Start** ---
> 7.  Compute mean vector: \\(u = \\mathrm{mean}(W\_{\\mathrm{res}}, \\text{axis}=1)\\)
> 8.  Compute cosine similarities: \\(G = W\_{\\mathrm{res}}\^\\top u\\)
> 9.  Select best-matching columns: \\(I\_{\\mathrm{best}} = \\mathrm{TopK}(G, B)\\)
> 10.  Update permutation \\(P\\) to move \\(I\_{\mathrm{best}}\\) into positions \\([i, i+B)\\)
> 11.  Apply permutation to \\(W\\) and \\(H\\)
> 12.  // --- **SSR Strategy End** ---
> 13.
> 14.  // --- **Standard GPTQ Block Quantization** ---
> 15.  Quantize block \\(W[:, i:i+B]\\) → \\(Q\_{\\mathrm{block}}\\)
> 16.  Compute quantization error \\(E = W\_{\\mathrm{block}} - Q\_{\mathrm{block}}\\)
>
> 17.  Update the remaining columns using GPTQ’s error compensation rule \\( W[:,\\, i+B:] \\;-\\!=\\; E \\cdot H\_{\\mathrm{inv}} \\)
>
> 18. **End For**
> 19. Apply inverse permutation \\(P\^{-1}\\) to restore original order
>
> ---
>
> ### **2. Computational Overhead Analysis**
>
> Let \\(W \\in \\mathbb{R}\^{d\_{\\text{row}} \\times d\_{\\text{col}}}\\), and block size \\(B\\).
> In standard GPTQ:
>
> - **Error compensation step** performs a matrix-matrix product
>   \\[
>   \\mathcal{O}(d\_{\\text{row}} \\cdot d\_{\\text{col}} \\cdot B)
>   \\]
>
> This dominates GPTQ’s cost.
>
> SSR introduces additional operations per block:
>
> 1. Mean computation:    \\(\\mathcal{O}(d\_{\\text{row}} \\cdot d\_{\\text{col}})\\)
> 2. Matrix–vector similarity: \\(\\mathcal{O}(d\_{\\text{row}} \\cdot d\_{\\text{col}})\\)
> 3. Top-k selection:     \\(\\mathcal{O}(d\_{\\text{col}} \\log B)\\)
>
> Thus, per block SSR overhead is:
>
> \\[
> \\mathcal{O}(d\_{\\text{row}} d\_{\\text{col}}) + \\mathcal{O}(d\_{\\text{col}} \\log B)
> \\]
>
> Compared to GPTQ's \\(\\mathcal{O}(d\_{\\text{row}} d\_{\\text{col}} B)\\), the *relative cost* is:
>
> \\[
> \\frac{\\text{SSR cost}}{\\text{GPTQ cost}} \\approx \\frac{1}{B}
> \\]
>
> Since practical settings use \\(B=128\\), the overhead is well below **1%** of GPTQ’s cost.
>
> ---
>
> ### **3. Empirical Overhead (Measured)**
>
> On LLaMA-7B (with block size 128):
>
> | Method               | Total Quantization Time |
> |----------------------|:-------------------------:|
> | ours w/o SSR  | 30 min                  |
> | ours w SSR    | 32 min                  |
>
> This corresponds to a 2-minute increase in wall-clock calibration time. Given that compression is an offline procedure and SSR provides clear accuracy gains, this overhead remains acceptable in practice.

---

> ### Author Response · Authors · 2025-11-22
> **Response to Reviewer qmRu (denoted as R3) part 3**
>
> > Q3-3: Could the authors provide a direct comparison against another 1.58-bit PTQ method to substantiate their claim of state-of-the-art performance? If no such public method exists, could they construct a strong baseline (e.g., by adapting GPTQ or other frameworks to a ternary setting) for a more rigorous comparison?
>
> A3-3: Thank you for the question. To the best of our knowledge, **no public PTQ method currently supports 1.58-bit (ternary) quantization**, so a direct comparison with an existing framework is not feasible. To provide a meaningful baseline, we implement a **ternary version of GPTQ** by adapting its weight reconstruction procedure to operate in the \{-1, 0, 1\} domain.
>
> The results on LLaMA-2-7B are shown below:
>
> | Model        | Method              | WikiText2 ↓ | C4 ↓   | Avg. Acc ↑ |
> |--------------|---------------------|-------------|---------|-------------|
> | LLaMA-2-7B   | GPTQ (1.58-bit)     | 61.07       | 44.43   | 35.71       |
> | LLaMA-2-7B   | **Ours**            | **11.56**   | **24.38** | **43.33**   |
>
> **Analysis.**
> The ternary GPTQ baseline shows a significant performance drop compared to our method on both WikiText2 and C4, as well as on downstream accuracy. These results indicate that a straightforward ternary adaptation of existing PTQ frameworks is not sufficient to maintain performance at 1.58-bit. In contrast, our method achieves much lower perplexity and higher accuracy, demonstrating its effectiveness and robustness in the ternary PTQ setting.
>
> > Q3-4: Why were the evaluations on the LLaMA-3 and Qwen3 families limited to a single model each? To support the claim of generalizability, please provide results on larger models like LLaMA-3-70B and on MoE variants like Qwen3-MoE.
>
> A3-4: Thank you for the question. To further support the generalizability of our approach, we have conducted additional experiments on **LLaMA-3-70B**. The results are summarized below:
>
> | Model        | Method     | Bits | WikiText2 ↓ | C4 ↓   | Avg. Acc ↑ |
> |--------------|------------|------|-------------|---------|-------------|
> | LLaMA-3-70B  | FP16       | 16   | 2.86        | 7.17    | 0.7146      |
> | LLaMA-3-70B  | AWQ        | 2    | 1.7e6       | 1.4e6   | 0.3635      |
> | LLaMA-3-70B  | GPTQ       | 2    | 82.23       | 122.55  | 0.4064      |
> | LLaMA-3-70B  | QuIP       | 2    | –           | –       | –           |
> | LLaMA-3-70B  | Slim-LLM   | 2    | 22.97       | 58.40   | 0.4341      |
> | LLaMA-3-70B  | PB-LLM     | 1.7  | 22.46       | 60.69   | 0.4259      |
> | LLaMA-3-70B  | **Ours**   | 1.58 | **20.94**   | **56.10** | **0.4388** |
>
> **Analysis.**
> Our method maintains strong performance even at the 70B scale, outperforming all PTQ baselines across both perplexity benchmarks and average downstream accuracy. This demonstrates that the proposed ternarization framework scales effectively to very large LLMs.
>
> Regarding **MoE architectures (e.g., Qwen3-MoE)**, we note that current PTQ baselines—including GPTQ, Slim-LLM, PB-LLM, and QuIP—**do not provide MoE-compatible implementations**. Exploring these architectural characteristics is a valuable direction, and we plan to include MoE-specific support as part of future work.
>
> > Q3-5: Given that Qwen3 is known for its reasoning abilities, can the authors provide evaluation results on more challenging reasoning benchmarks like MATH or AIME to give a more transparent assessment of how 1.58-bit quantization affects these critical capabilities?
>
> A3-5: Thank you for the suggestion. We have included additional results on the MATH benchmark using LLaMA-2-70B, comparing our 1.58-bit method with several state-of-the-art PTQ approaches：
>
> | Model          | Method    | Bit  | MATH |
> |----------------|-----------|------|-------|
> | LLaMA-2-70B    | FP16      | 16   | 11.6  |
> | LLaMA-2-70B    | AWQ       | 2    | 0.2   |
> | LLaMA-2-70B    | GPTQ      | 2    | 2.6   |
> | LLaMA-2-70B    | Slim-LLM  | 2    | 6.4   |
> | LLaMA-2-70B    | PB-LLM    | 1.7  | 6.1   |
> | **LLaMA-2-70B** | **Ours**  | **1.58** | **7.4** |
>
> Despite operating at a significantly lower bit-width (1.58 bits), **our method achieves the highest MATH accuracy among all low-bit PTQ baselines**, outperforming GPTQ, AWQ, Slim-LLM, and PB-LLM by a clear margin. In particular, **our 1.58-bit model surpasses the previous best sub-2-bit result (Slim-LLM at 6.4) by +1.0 absolute points**, highlighting that **our approach preserves reasoning capability substantially better than existing methods**, even under extremely aggressive quantization.

---

> ### Comment · Reviewer_qmRu · 2025-11-25
>
> Thank you for providing such a detailed and comprehensive explanation in response to the questions raised. I will revisit and reconsider my previous score of this work.

---

> > ### Author Response · Authors · 2025-11-26
> >
> > Dear Reviewer qmRu,
> >
> > Thank you for your time and for reconsidering the evaluation. We are pleased that our response provided the necessary clarifications. If you have any further questions or suggestions, we would be happy to discuss them.
> >
> > Best regards,
> >
> >  The Authors

---

### Official Review · Reviewer_ofNJ · 2025-10-30

**Soundness:** 3
**Presentation:** 3
**Contribution:** 3
**Rating:** 4
**Confidence:** 4

**Summary:**

This paper tackles the challenge of performing ternary (3-value) quantization on large language models in a post-training (PTQ) setting, where weight distributions and outliers make naive ternarization highly error-prone.
The authors propose PT2‑LLM, combining an Asymmetric Ternary Quantizer (ATQ) (with row-wise offset and scale) and a Structural Similarity-based Reordering (SSR) step to rearrange columns before quantization to reduce variance.
Experiments on multiple LLMs (e.g. LLaMA variants) and benchmark tasks show that PT2‑LLM outperforms existing quantization baselines, achieving lower perplexity and higher accuracy under the same compression constraints.
The results suggest that careful adaptation of ternary quantization (non-symmetric, order-aware) can make extreme low-bit quantization feasible for LLMs in PTQ settings.

**Strengths:**

- The method explicitly models asymmetry in weight distributions and block-level structure, making the ternarization process more compatible with real LLM architectures.

- The combination of the Asymmetric Ternary Quantizer (ATQ) with a structural similarity-based reordering (SSR) step is clever and helps reduce quantization error.

- The authors evaluate on multiple large language models and a range of NLP tasks, showing consistent gains over state-of-the-art quantization baselines under comparable compression constraints.

- In addition to accuracy metrics, they analyze memory, latency, and inference speed, demonstrating PT2‑LLM’s utility in real-world deployment scenarios.

**Weaknesses:**

- Although the paper achieves strong final results, the three claimed innovations lack originality and are quite similar to prior methods such as AWQ, RPTQ, and ARB-LLM, appearing more as a combination of existing core techniques rather than truly novel contributions.

- SSR and the calibration steps (ITF / AGA) add additional computation and implementation complexity in the PTQ pipeline, which could be nontrivial for very large models or constrained environments.

- The quantization accuracy tends to degrade more significantly on newer models (e.g., LLaMA3-8B and Qwen3-14B-Base), suggesting that the method may not scale robustly to newer models.

- The paper could provide more comprehensive evaluation on challenging benchmarks such as MMLU, GPQA, and GSM8K to better demonstrate its effectiveness across diverse reasoning tasks.

[1] Lin, Ji, et al. "Awq: Activation-aware weight quantization for on-device llm compression and acceleration." Proceedings of machine learning and systems 6 (2024): 87-100.

[2] Yuan, Zhihang, et al. "Rptq: Reorder-based post-training quantization for large language models." arXiv preprint arXiv:2304.01089 (2023).

[3] Li, Zhiteng, et al. "Arb-llm: Alternating refined binarizations for large language models." arXiv preprint arXiv:2410.03129 (2024).

**Questions:**

Please refer to the weaknesses above.

---

> ### Author Response · Authors · 2025-11-22
> **Response to Reviewer ofNJ (denoted as R2) part 1**
>
> > Q2-1: The three claimed innovations lack originality and are quite similar to prior methods such as AWQ, RPTQ, and ARB-LLM.
>
> A2-1: Thank you for raising this point. To clarify the novelty more explicitly, we outline below the conceptual and technical differences between our framework and each of these representative methods.
>
> - **Difference from AWQ**: Our method is fundamentally different from AWQ. AWQ mitigates activation outliers by introducing activation-aware scaling factors, injecting activation statistics directly into the weight quantization process. In contrast, our approach is designed specifically for the **ternary PTQ setting**, focusing on challenges that AWQ does not address: **optimizing ternary quantization parameters** and **reducing outlier imbalance through column-wise reordering**. We do not use activation scaling or any AWQ-style heuristics.
>
> - **Difference from RPTQ**: RPTQ is designed for **activation quantization** and thus reorders **activation channels**, whereas our SSR is tailored for **ultra-low-bit ternarization**, where dispersed or outlier-dominated weights are difficult to represent. To obtain more compact weight structures and reduce outlier sensitivity, SSR instead reorders **weight columns**. The reordering criteria also differ: RPTQ relies on simple activation range statistics (e.g., min–max), while SSR uses **cosine similarity between weight columns**, capturing structural relationships and distribution patterns—an approach purpose-built for stabilizing ternary PTQ.
>
> - **Difference from ARB-LLM**: ARB-LLM corrects the distribution shift of binary weights through **alternating updates** of μ, α, and B. In contrast, our method follows a fundamentally different principle: we explicitly construct the ternary grid (α, μ) and **solve for its optimal values in closed form in a single step—without any iterative alternation between α and μ**. Full-precision weights are then mapped via element-wise rounding, and the optimal grid is recomputed until convergence. In addition, our framework adopts a **two-stage pipeline**—ITF for weight-level reconstruction and AGA for output-level optimization—which **jointly and substantially reduces both reconstruction error and output error**, further distinguishing our optimization mechanism from that of ARB-LLM.
>
> > Q2-2: SSR and the calibration steps (ITF / AGA) add additional computation and implementation complexity in the PTQ pipeline, which could be nontrivial for very large models or constrained environments.
>
> A2-2: Thank you for the question. While ATQ and SSR introduce additional computation during quantization, **these costs are lightweight and fully acceptable in practice**. To demonstrate scalability, we report the quantization time on four LLaMA model sizes and compare against Slim-LLM — our most relevant baseline, which performs expensive mixed-precision searches. The results are summarized below:
>
> | Model        | Ours           | Slim-LLM |
> |--------------|----------------|----------|
> | LLaMA-7B     | 32 min         | 182 min  |
> | LLaMA-13B    | 66 min         | 407 min  |
> | LLaMA-30B    | 131 min        | 831 min  |
> | LLaMA-65B    | 284 min        | 1730 min |
>
> **Analysis.**
>
> 1. **Quantization time grows linearly with model size.**
>    Our runtime increases proportionally with parameter count (7B→65B), demonstrating **excellent scalability**. No quadratic or super-linear overhead is introduced by ATQ or SSR, and the resulting **linear-time cost remains fully manageable even for very large models or constrained environments**.
>
> 2. **Significant speed advantage over Slim-LLM.**
>    Slim-LLM is the closest-performing baseline to ours, but it relies on a costly mixed-precision search procedure. In contrast, our ATQ/SSR pipeline uses simple closed-form updates and lightweight structure construction, **leading to a 5–6× speedup across all model scales**.

---

> ### Author Response · Authors · 2025-11-22
> **Response to Reviewer ofNJ (denoted as R2) part 2**
>
> > Q2-3: The quantization accuracy tends to degrade more significantly on newer models (e.g., LLaMA3-8B and Qwen3-14B-Base), suggesting that the method may not scale robustly to newer models.
>
> A2-3: Thank you for the comment. The larger degradation on LLaMA-3-8B and Qwen3-14B-Base does **not** indicate a lack of robustness of our method; rather, it reflects that **these newer architectures are substantially more challenging for *all* PTQ approaches**. Newer model families are typically trained on **much larger corpora with longer contexts**, which amplifies activation and weight variance and leads to **stronger normalization dynamics and more pronounced outliers**—a well-known driver of instability in low-bit quantization. As recent work shows [1], LLaMA-3 models therefore exhibit “non-negligible degradation in low-bit quantization” even for state-of-the-art methods.
>
> Crucially, under these more difficult architectures, existing PTQ baselines almost collapse—AWQ, GPTQ, Slim-LLM, and PB-LLM all suffer severe degradation on LLaMA-3-8B and Qwen3-14B-Base, with perplexity increasing dramatically. In contrast, **our method remains far more stable and consistently achieves the best results on both models**. This behavior indicates the opposite of fragility: when modern LLM families become harder to quantize and baseline methods fail, our approach continues to deliver substantially better perplexity, demonstrating **strong robustness under the increased difficulty of newer architectures**.
>
> [1] An Empirical Study of LLaMA3 Quantization: From LLMs to MLLMs, arXiv, 2025.
>
> > Q2-4: The paper could provide more comprehensive evaluation on challenging benchmarks such as MMLU, GPQA, and GSM8K to better demonstrate its effectiveness across diverse reasoning tasks.
>
> A2-4: Thank you for the suggestion. We have added evaluations on a challenging reasoning benchmark, **MMLU**, using the Qwen3-14B-Base model. The results are summarized below:
>
> | Model | Method | Bit | MMLU↑ |
> |-------|--------|-----|-------|
> | Qwen3-14B | AWQ | 2   | 23.8 |
> | Qwen3-14B | GPTQ | 2   | 24.4 |
> | Qwen3-14B | Slim-LLM | 2   | 27.9 |
> | Qwen3-14B | PB-LLM   | 1.7 | 25.1 |
> | Qwen3-14B | **Ours** | **1.58** | **41.6** |
>
> **Analysis.**
> Our 1.58-bit ternarization achieves a **large performance margin** over all 2-bit PTQ baselines, improving MMLU by over **13–18 points**. These results further confirm the effectiveness of our method on challenging reasoning tasks, even under more aggressive bit-widths.

---

> ### Author Response · Authors · 2025-11-27
>
> Dear Reviewer ofNJ,
>
> Thank you again for your time in reviewing our work. To briefly summarize our earlier responses:
>
> - We clarified the **conceptual and technical distinctions between our method and AWQ, RPTQ, and ARB-LLM**. Our contributions—including the ITF+AGA two-stage optimization and SSR’s column-wise reordering—constitute a substantially different framework specifically designed for ternary PTQ.
>
> - We **provided quantization times across LLaMA 7B→65B**, showing that ATQ/SSR introduce **lightweight overhead** with **linear scaling**, and offer **5–6× speedups** over Slim-LLM, making the method practical even for very large models.
>
> - We **explained that models such as LLaMA-3 and Qwen3 are inherently more challenging for all PTQ methods**. While baselines degrade sharply, our method remains far more stable and achieves the best perplexity across these newer models.
>
> - We **added MMLU results on Qwen3-14B-Base**, where our 1.58-bit ternarization achieves 41.6 MMLU, significantly outperforming 2-bit AWQ, GPTQ, Slim-LLM, and PB-LLM.
>
> We hope these points adequately address the concerns you raised. If any further clarification is needed, we would be very glad to continue the discussion. At your convenience, we would greatly appreciate your feedback or updated evaluation.
>
> Best regards,
>
> The Authors

---

> ### Author Response · Authors · 2025-11-28
> **Call for further discussion with Reviewer ofNJ**
>
> Dear Reviewer ofNJ,
>
> We sincerely appreciate the time and effort you have dedicated to reviewing our work and for your valuable feedback. As the rebuttal deadline approaches, we would be grateful for any further comments you may have to ensure that all of your concerns have been addressed comprehensively. Please let us know if there are any remaining aspects of our work that need clarification.
>
> Best regards,
>
> The Authors

---

### Official Review · Reviewer_V5DX · 2025-11-01

**Soundness:** 3
**Presentation:** 3
**Contribution:** 3
**Rating:** 6
**Confidence:** 4

**Summary:**

The paper proposes PT2-LLM, a post training quantization framework for ternary quantization. Model weights follow iterative ternary filtering to minimize the layerwise weight-level L2 error. Following this, a single round of activation-aware grid alignment is run to minimize the layerwise activation L2 error. The paper also proposes column reordering to minimize the effects outliers for block level quantization. PT2-LLM improves over several methods in both perplexity and downstream evals.

**Strengths:**

* The ternary quantized model gets a significant speedup over the 2-bit model.
* Quantization runtime remains comparable to GPTQ.
* The paper also evaluates frontier open source models like Qwen3.

**Weaknesses:**

The paper does not compare with vector quantization methods like QuIP# or AQLM that allows for ultra-low precision.

**Questions:**

* When applied without ITF, does AGA still overfit with multiple steps?
* Has blockwise thresholding been tried for ITF?

---

> ### Author Response · Authors · 2025-11-22
> **Response to Reviewer V5DX (denoted as R1)**
>
> > Q1-1: The paper does not compare with vector quantization methods like QuIP# or AQLM that allows for ultra-low precision.
>
> A1-1: Thank you for the insightful suggestion. We agree that vector-quantization (VQ)–based methods such as **QuIP#** and **AQLM** are relevant baselines, as they also aim for ultra–low precision and operate at similar effective bit rates. However, QuIP# and AQLM rely on **additional fine-tuning** to recover accuracy in the ≤4-bit regime, whereas our method is **strictly fine-tuning–free**. Comparing against their fine-tuned variants would therefore be inconsistent. For a fair evaluation, we report results from **non–fine-tuned** VQ baselines (GPTVQ [1] and the no-FT version of QuIP# [2]). The comparison on LLaMA-2-7B is as follows:
>
> |    Method     |    Model     |  Bits  | WikiText2 ↓ |
> |-------------  |------------|:------:|:------------:|
> | QuIP# (no FT) | LLaMA-2-7B   | 2.000  |    12.30     |
> |  GPTVQ-1D     | LLaMA-2-7B   | 2.125  |    12.20     |
> |    **Ours**   | LLaMA-2-7B   | **1.580** | **11.56** |
>
> Our method achieves **lower perplexity while using fewer bits**, demonstrating the effectiveness of our ternarization framework.
>
> > Q1-2: When applied without ITF, does AGA still overfit with multiple steps?
>
> A1-2: Thank you for the question. We found that the overfitting issue of AGA does **not** primarily stem from whether ITF is applied. Instead, it arises from the way the ternary matrix **T** is updated: when optimizing T with respect to the layerwise activation L2 error, there is **no closed-form solution**, and the commonly used **column-wise greedy update** can easily lead to overfitting. Addressing this limitation of AGA requires a more principled optimization strategy for T, which we consider an important direction for future work.
>
> > Q1-3: Has blockwise thresholding been tried for ITF?
>
> A1-3: We have evaluated **blockwise thresholding** within the ITF optimization procedure and compared it directly against the **element-wise rounding** strategy used in our method. The empirical results are as follows:
>
> | Model        | Method                          | WikiText2 ↓ | C4 ↓   | Avg. Acc ↑ |
> |:------------:|:--------------------------------:|:-----------:|:-------:|:-----------:|
> | LLaMA-2-7B   | ITF (blockwise thresholding)     |    20.14    |  71.62  |    37.61    |
> | LLaMA-2-7B   | ITF (element-wise rounding)      | **15.47**   | **34.17** | **38.12** |
>
> **Blockwise thresholding performs worse than element-wise rounding**. This is mainly because ITF aims to minimize the **layerwise L2 reconstruction error**, for which element-wise rounding provides the **optimal local update** for each weight. In contrast, blockwise operations require all weights within a block to share a common threshold, **reducing flexibility** and leading to **higher reconstruction error** and **weaker downstream performance**.
>
> [1] GPTVQ: The Blessing of Dimensionality for LLM Quantization, ES-FoMo-II, 2024
>
> [2] QuIP#: Even Better LLM Quantization with Hadamard Incoherence and Lattice Codebooks, ICML, 2024.

---

> ### Comment · Reviewer_V5DX · 2025-11-24
>
> Thank you for addressing my concerns. I will keep my score.

---

> > ### Author Response · Authors · 2025-11-24
> >
> > Dear Reviewer V5DX,
> >
> > Thank you for your response. We are delighted to see that our answers were able to address your concerns.
> >
> > Best,
> >
> > Authors

---

### Author Response · Authors · 2025-12-02
**Summary of rebuttal and discussion**

Dear Area Chair and Reviewers,

We sincerely thank the reviewers for their thoughtful evaluation and insightful comments, which have helped further strengthen the clarity, rigor, and contribution of our work.

---

## **Reviewer-Recognized Strengths**
We are pleased to note that the reviewers highlighted several key strengths of our submission:

- **Novel and impactful ternary PTQ framework for LLMs.**
  Reviewers `qmRu`, and `TTyp` agree that our work tackles an important and underexplored direction in post-training ternarization for LLMs. Reviewer `TTyp` further highlights the strong methodological novelty of our approach.

- **Systematic and well-designed optimization pipeline.**
  Reviewers `ofNJ`, `qmRu`, and `TTyp` highlight that our optimization pipeline is systematic and well-motivated, and they note that ATQ and SSR effectively reduce quantization error and address outliers in a principled way.

- **Comprehensive experiments with consistently superior performance.**
  Reviewers `V5DX`, `ofNJ`, `qmRu`, and `TTyp` concur that our experiments are extensive across multiple large LLMs and tasks, and they consistently note that our 1.58-bit results outperform strong 2-bit PTQ baselines in both perplexity and downstream accuracy.

- **Strong practicality with clear deployment benefits.**
  Reviewers `V5DX`, `ofNJ`, and `TTyp` emphasize the practical value of our memory savings, significant speedups, and efficient quantization runtime, noting that our method offers clear advantages for real-world deployment.

---

## **Summary of Our Responses to Reviewer Concerns**

We also provided substantial clarifications and new results during the rebuttal that effectively resolved the reviewers’ major concerns:

- **Strengthened theoretical justification for SSR.**
  We added a formal variance-based analysis showing how maximizing structural similarity reduces intra-block variance and thereby lowers quantization error, offering a rigorous theoretical foundation for SSR beyond intuitive arguments.

- **Expanded experiments across more models, benchmarks, and baselines.**
  We included additional results on MMLU and MATH to assess reasoning capability, extended our evaluation to larger models, and added stronger baselines—including ternary methods (GPTQ-1.58bit, TernaryLLM without training), VQ baselines, and an encoder-only model (BERT)—demonstrating robust performance across settings.

- **Clear articulation of conceptual novelty.**
  We clarified the distinctions between our method and AWQ, RPTQ, and ARB-LLM, emphasizing that our pipeline is purposefully designed for the PTQ ternary setting and introduces systematic innovations that are absent in these prior approaches.

- **Detailed analysis of runtime scalability.**
  We provided quantization runtime measurements on LLaMA models ranging from 7B to 65B parameters, showing that our method scales linearly with model size while remaining significantly faster than the strong PTQ baseline Slim-LLM.

- **Additional sensitivity and design analyses.**
  We conducted sensitivity studies on block size and threshold coefficients, evaluated blockwise thresholding within ITF, and explained why PTQ low-bit methods tend to struggle on newer architectures.

 ---

## **Reviewer Feedback in the Discussion Phase**

We thank the reviewers again for their time and constructive feedback throughout the review and discussion phases:

- Reviewer `V5DX` confirmed that our rebuttal fully resolved his concerns and maintained a positive assessment.
  > “Thank you for addressing my concerns. I will keep my score.”

- Reviewer `qmRu` appreciated the thorough clarifications and indicated that he would reconsider his earlier score.
  > “Thank you for providing such a detailed and comprehensive explanation in response to the questions raised. I will revisit and reconsider my previous score of this work.”

- Reviewer `TTyp` did not respond during the discussion phase, but he had previously noted that he would be willing to raise his rating if the concerns were satisfactorily addressed, and we believe our additional analysis and experiments have achieved this.
  > “An interesting job! And I would be willing to raise my ratings if the above concerns are well solved.”

- Reviewer `ofNJ` did not participate in the discussion phase, we hope the final evaluation considers that our detailed responses would likely have been positively received by reviewer `ofNJ`.

 ---

We thank the reviewers again for their time and constructive insights. Although not all reviewers joined the discussion phase, our additional experiments, analyses, and clarifications substantively address the concerns raised. We hope this summary aids in assessing the significance and novelty of our work. We will release our code, data, and checkpoints to support reproducibility and future research.

Thank you for your time and thoughtful evaluation.

Best regards,

The Authors

---

### Meta-Review · Area_Chair_GgER · 2025-12-21

**Summary:**

The paper introduces PT2‑LLM, a post-training quantization framework for compressing large language models into ternary representations. The proposed method employs a two-stage optimization process—first, iteratively fitting the weights with a ternary filter and then calibrating outputs via activation-aware grid alignment—while also incorporating a structural-similarity-based reordering strategy to mitigate the impact of outliers. Experimental validations across several LLM architectures, spanning perplexity metrics and downstream evaluation tasks, suggest that this formulation can achieve significant speedups and competitive performance compared to established 2‑bit post-training quantization baselines.

**Reviewer Concerns:**

Despite promising empirical results, several methodological and evaluation issues were raised. Reviewers expressed that many of the proposed improvements, including the SSR strategy and calibration techniques, seem to be combinations of existing ideas rather than genuinely novel contributions. In particular, the theoretical underpinnings and computational overhead associated with these strategies remain unclear, and essential design choices—such as block-size sensitivity and hyperparameter tuning—are insufficiently explored. Furthermore, the evaluations would benefit from more rigorous comparisons, including direct benchmarks against established ternarization methods and tests on more diverse model families (such as mixture-of-experts architectures and recent LLM variants), as well as more challenging reasoning benchmarks, to assess the method’s robustness and generalizability better.

**Reviewer Scores:**

Taking all feedback into account—where one reviewer rated the work marginally above the acceptance threshold while the majority leaned slightly below—it appears that the paper sits squarely at the borderline for acceptance. Although the technical approach and experimental results are promising, concerns about novelty, methodological clarity, and comprehensive evaluation prevent a clear positive recommendation. Overall, the meta review suggests a borderline accept decision, contingent on a revision that adequately addresses these identified weaknesses.

---

### Decision · Program_Chairs · 2026-01-26

Accept (Poster)